# Loss of the transcription factor *Meis1* prevents sympathetic neurons target-field innervation and increases susceptibility to sudden cardiac death

Fabrice Bouilloux[1†], Jérôme Thireau[2†], Stéphanie Ventéo[1], Charlotte Farah[2], Sarah Karam[2], Yves Dauvilliers[3], Jean Valmier[1], Neal G Copeland[4], Nancy A Jenkins[4], Sylvain Richard[2], Frédéric Marmigère[1*]

[1] Institute for Neurosciences of Montpellier, Institut national de la santé et de la recherche médicale, Montpellier, France; [2] Physiologie et Médecine Expérimentale du cœur et des Muscles, INSERM U1046, CNRS UMR 9214, University of Montpellier, Montpellier, France; [3] Sleep Unit, Department of Neurology, Gui-de-Chauliac hospital, Montpellier, France; [4] Cancer Research Program, The Methodist Hospital Research Institute, Houston, United States

**Abstract** Although cardio-vascular incidents and sudden cardiac death (SCD) are among the leading causes of premature death in the general population, the origins remain unidentified in many cases. Genome-wide association studies have identified Meis1 as a risk factor for SCD. We report that *Meis1* inactivation in the mouse neural crest leads to an altered sympatho-vagal regulation of cardiac rhythmicity in adults characterized by a chronotropic incompetence and cardiac conduction defects, thus increasing the susceptibility to SCD. We demonstrated that Meis1 is a major regulator of sympathetic target-field innervation and that *Meis1* deficient sympathetic neurons die by apoptosis from early embryonic stages to perinatal stages. In addition, we showed that Meis1 regulates the transcription of key molecules necessary for the endosomal machinery. Accordingly, the traffic of Rab5[+] endosomes is severely altered in Meis1-inactivated sympathetic neurons. These results suggest that *Meis1* interacts with various trophic factors signaling pathways during postmitotic neurons differentiation.

*For correspondence: frederic.marmigere@inserm.fr

[†] These authors contributed equally to this work

Competing interests: The authors declare that no competing interests exist.

## Introduction

Cardio-vascular diseases and sudden cardiac death (SCD) are together among the highest causes of mortality in the general population, and ventricular arrhythmias are identifiable in almost all cardiac diseases. The incidence of SCD and the poor outcome of sudden cardiac arrest make heart diseases a leading cause of mortality in young individuals (*Meyer et al., 2012*). Among frequent causes of SCD, numerous genetic mutations have been identified as responsible for arrhythmogenic cardiomyopathies, cardiac malformations or sympatho-vagal dysfunctions (*Basso et al., 2012*; *Chopra and Knollmann, 2011*; *Fukuda et al., 2015*). Congenital cardiac innervation defects or dysfunction are largely implicated in SCD and often involve molecules that are essential for the developmental guidance and axonal growth of cardiac sympathetic nerves, such as Sema3a and Ngf (*Dae et al., 1997*; *Fukuda et al., 2015*). Recently, two independent genome wide association studies based on abnormal cardiac conduction as an increased susceptibility to SCD have emphasized a restricted number of genes among which *Meis1* (*Pfeufer et al., 2010*; *Smith et al., 2011*). Meis1 is a transcription factor of the TALE homeobox gene family, known to play important roles during embryonic

**eLife digest** Nerve cells called sympathetic neurons can control the activity of almost all of our organs without any conscious thought on our part. For example, these nerve cells are responsible for accelerating the heart rate during exercise.

In a developing embryo, there are initially more of these neurons than are needed, and only those that develop correctly and form a connection with a target cell will survive. This is because the target cells provide the growing neurons with vital molecules called neurotrophins, which are trafficked back along the nerve fiber and into the main body of the nerve cell to ensure its survival. However, it is largely unknown which proteins or genes are also involved in this developmental process.

Now, Bouilloux, Thireau et al. show that if a gene called *Meis1* is inactivated in mice, the sympathetic neurons start to develop and grow nerve fibers, but then fail to establish connections with their target cells and finally die. The *Meis1* gene encodes a transcription factor, which is a protein that regulates gene activity. Therefore, Bouilloux, Thireau et al. looked for the genes that are regulated by this transcription factor in sympathetic neurons. This search uncovered several genes that are involved in the packaging and trafficking of molecules within cells.

Other experiments then revealed that the trafficking of molecules back along the nerve fiber was altered in mutant neurons in which the *Meis1* gene had been inactivated. Furthermore, *Meis1* mutant mice had problems with their heart rate, especially during exercise, and an increased risk of dying from a sudden cardiac arrest.

These findings reveal a transcription factor that helps to establish a connection between a neuron and its target, and that activates a pattern of gene expression that works alongside the neurotrophin-based signals. Since all neurons undergo similar processes during development, future work could ask if comparable patterns of gene expression exist in other types of neurons, and if problems with such processes contribute to some neurodegenerative diseases.

development, cardiogenesis and postnatal cardiomyocytes turnover (*Moens and Selleri, 2006*; *Stankunas et al., 2008*; *Mahmoud et al., 2013*).

The early steps of sympathetic neurons development have been extensively documented. During sympathetic neurons formation, a concerted cross-regulated transcriptional network induced by BMPs and involving among others the transcription factors Hand2, Ascl1, Phox2b and Gata3 initially instructs neural crest cells to a sympathetic fate and a noradrenergic phenotype by up-regulating the expressions of general neuronal markers and the monoamines biosynthesis enzymes tyrosine hydroxylase (TH) and dopamine-β-hydroxylase (DBH) (*Rohrer, 2011*). Subsequently, connecting sympathetic neurons with their peripheral target relies on combined actions of locally secreted ligands of various families of trophic factors (*Glebova and Ginty, 2005*; *Ernsberger, 2008*; *2009*). Artn, a growth factor of the Gdnf family of ligands, acting on GFRα3 and Ret, and the neurotrophin Ntf3 are initially involved in proximal axonal growth, and members of the endothelin family serve as vascular-derived guidance cues for sympathetic neurons (*Andres et al., 2001*; *Enomoto et al., 2001*; *Honma et al., 2002*; *Kuruvilla et al., 2004*; *Makita et al., 2008*; *Manousiouthakis et al., 2014*), allowing sympathetic axons to reach the vicinity of target organs. At later stages, distal axonal growth and target-field innervation are under the control of retrograde Ntrk1/Ngf signaling for most organs including the heart (*Glebova and Ginty, 2004*). Ngf is secreted in limiting amounts by target organs and binds the high affinity receptor Ntrk1 to induce its phosphorylation and internalization both by clathrin-dependent and independent mechanisms. Retrogradely transported Ngf/Ntrk1 is mediated by Rab5-positive endosomes. This signaling induces phosphorylation of CREB and allows the maintenance of Ntrk1 expression both of which are required to promote axonal growth and branching of sympathetic neurons within their peripheral target organs, and to protect them from naturally occurring cell death (*Crowley et al., 1994*; *Smeyne et al., 1994*; *Riccio et al., 1997,*; *1999*; *Kuruvilla et al., 2000*; *MacInnis et al., 2003*; *Ye et al., 2003*; *Kuruvilla et al., 2004*; *Valdez et al., 2005*; *Harrington et al., 2011*; *Suo et al., 2014*). Whereas part of the transcriptional program regulating the induction of Ntrk1 expression has been revealed in the peripheral nervous

system (PNS) through analysis of the effects of *Klf7* inactivation (*Lei et al., 2005*), the transcriptional regulation of molecules implicated in retrograde Ngf/Ntrk1 transport mechanisms in particular or endosomal transport in general are unknown.

We aimed in our study at investigating if Meis1 participates in neural aspects of cardiac regulation by intervening in the differentiation of sympathetic neurons and how *Meis1* mutations could mediate susceptibility for unexplained SCD.

In this work, we highlight that loss of *Meis1* function in early sympathetic neurons leads to an imbalanced sympatho-vagal regulation of cardiac functions in adult mice resulting in increased susceptibility to SCD. We further demonstrate that this imbalance results from neurodevelopmental defects leading to sympathetic neurons apoptosis and we position *Meis1* as novel actor in sympathetic neurons differentiation by its role in distal target-field innervation. Among the target genes regulated by Meis1, we evidenced a transcriptional regulation of proteins necessary for retrograde transport, endosomes formation, functioning and traffic. Accordingly, following specific inactivation of *Meis1* in peripheral neurons, the traffic of Rab5$^+$ early endosomes is impaired and sympathetic target-field innervation is compromised.

## Results

### *Meis1* inactivation in the PNS compromises the sympatho-vagal regulation of cardiac function.

To examine the consequences of specific inactivation of *Meis1* in the mouse peripheral nervous system (PNS), the PLAT$^{CRE}$ strain was crossed with the *Meis1$^{LoxP/LoxP}$* strain (*Figure 1—figure supplement 1A*). Immunochemistry and Western blot analysis showed a complete loss of *Meis1* in sympathetic neurons from E16.5 embryo and adult mutant mice (*Figure 1A*). When grown in a C57BL/6 genetic background, *PLAT$^{CRE}$::Meis1$^{LoxP/LoxP}$* mice carried *Meis1* alleles according to a Mendelian ratio but newborn pups died within the first days following birth. When grown in a mixed genetic background, 50% of the mutant mice survived beyond early neonatal stages and exhibited blepharoptosis (*Figure 1A*), a physiological sign of sympathetic neurons dysfunction. However, most mutants died prematurely and unpredictably before reaching adult stages (*Figure 1—figure supplement 1B*). The premature death of *PLAT$^{CRE}$::Meis1$^{LoxP/LoxP}$* mice together with their drooping eyelids phenotype led us to hypothesize impairments of sympathetic functions and consequent SCD. Accordingly, we found that newborn and adult mutants exhibited a dramatic reduction in cardiac sympathetic innervation by TH$^+$ fibers (*Figure 1A* and *Figure 1—figure supplement 1F and G*). WT and *PLAT$^{CRE}$::Meis1$^{LoxP/LoxP}$* adult mice were telemetrically monitored for ECG recordings and processed with Heart Rate Variability (HRV) analysis, a method allowing quantifying autonomic nervous system (ANS) activity on heart rate. Measurements of the mean PR, QRS and QTc intervals on sinus cycle during baseline showed no differences between groups (*Figure 1—figure supplement 1C*), excluding the participation of intrinsic cardiac ion channels mediating conduction or repolarization in this phenotype. However, mutant mice developed paroxysmal abnormal atrial and/or atrioventricular nodal conduction that led to spontaneous bradycardia or desynchronization, in which P waves were either progressively lengthened, absent, premature or retrograde (*Figure 1B*). These altered conductions were concomitant with a high occurrence of sinus arrest (*Figure 1E*) and a permanent bradycardia (overall increased mean RR interval), corroborated by an increased HRV parameter SDNN (a calculation of total heart beat-to-beat variability resulting from cardiac ANS activity) (*Figure 1C*). HRV analysis in the frequency domain (calculated by Fast Fourier transform applied on exclusively sinus successive RR intervals) confirmed a severe decrease in LF/HF ratio (a marker of sympatho-vagal activity) due to a drop in the LF band (a marker of sympathetic activity) in *PLAT$^{CRE}$::Meis1$^{LoxP/LoxP}$* *vs* WT mice without affecting the HF component (a marker of parasympathetic activity), demonstrating that the sympathetic only but not the parasympathetic component is affected (*Figure 1C*). WT and mutant mice were challenged by i.p. injection of the sympatho-mimetic isoproterenol or by the muscarinic agonist carbamoylcholine chloride (*Figure 1D*). The increased and decreased RR intervals following carbamoylcholine chloride and isoproterenol injections respectively (*Figure 1—figure supplement 1D*) indicated that the cholinergic and β-adrenergic machineries within cardiomyocytes are functional in both animals groups. However, whereas isoproterenol induced-heart rate acceleration was rapidly and progressively counterbalanced by endogenous parasympathetic activation, carbamoylcholine chloride challenge caused a persistent

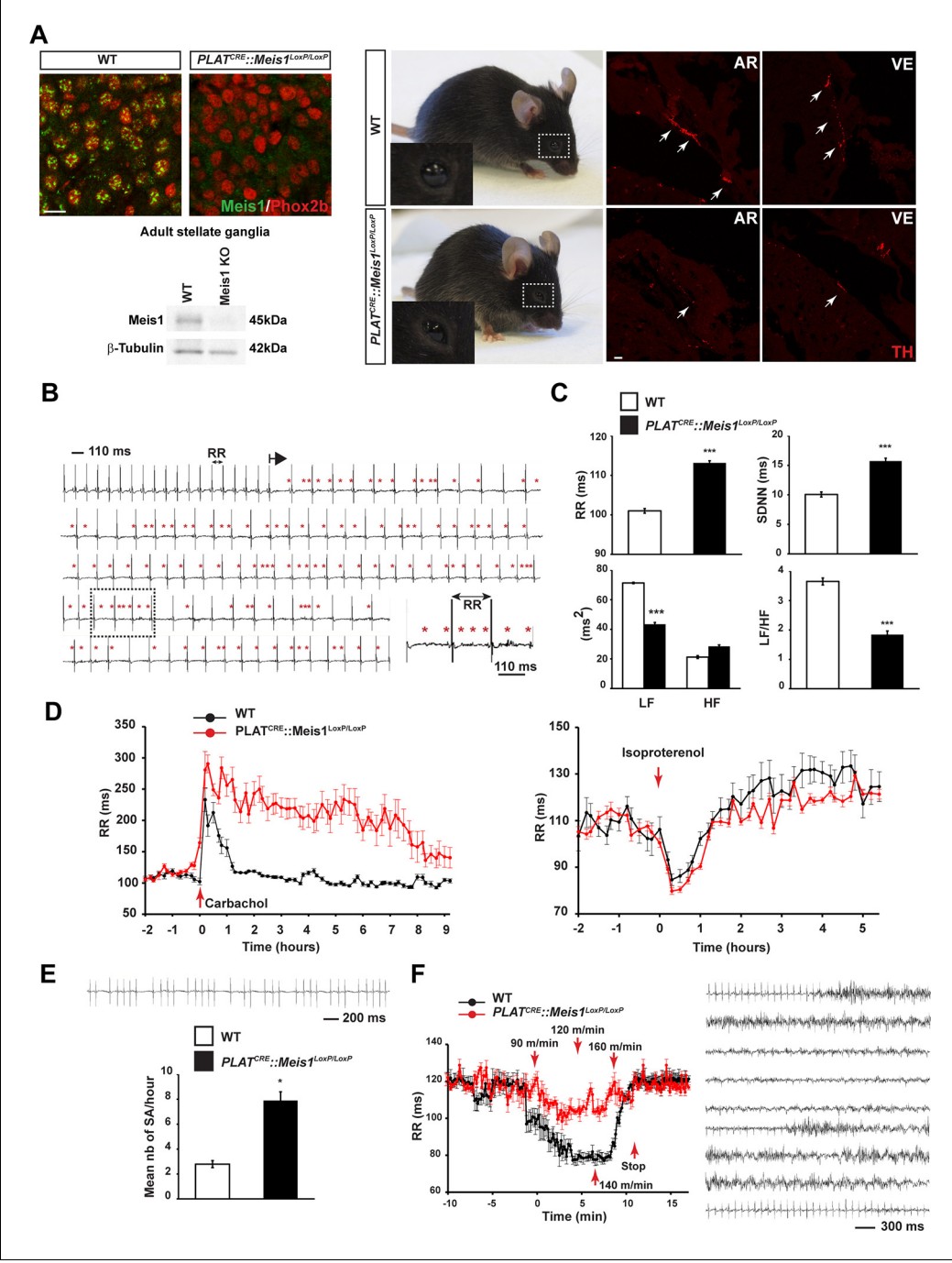

**Figure 1.** Specific *Meis1* inactivation in the PNS results in sympatho-vagal deregulation of cardiac functions. (**A**) Immunochemistry for Meis1 and the transcription factor Phox2b on sections of SCGs from E16.5 WT and *PLAT^CRE^:: Meis1^LoxP/LoxP^* mice, and western blot analysis for Meis1 from adult stellate ganglia. Surviving adult mutants exhibited blepharoptosis (white dotted square). Immunostaining for TH$^+$ sympathetic fibers in the heart of newborn WT and *PLAT^CRE^::Meis1^LoxP/LoxP^* mice. VE = Ventricle, AR = Atrium. White arrows point at TH$^+$ sympathetic fibers. Scale bar = 20 μm. (**B**) Example of an ECG trace recorded in an adult male *PLAT^CRE^::Meis1^LoxP/ LoxP^* mouse. The black arrow indicates the start of a junctional bradycardia and the red asterisks mark abnormal P waves locations. Dotted black square delineates the power magnification of the ECG trace showed. (**C**) HRV analysis of the ECGs recordings of *PLAT^CRE^::Meis1^LoxP/LoxP^* and WT mice. Graphical representations of the mean RR interval, the standard deviation of all normal to normal beats (SDNN), the power of low frequency band (LF), the power of the high frequency band (HF) and the LF/HF ratio. (**D**) Time course of the mean RR interval during isoproterenol or carbamoylcholine chloride challenges in WT and *PLAT^CRE^::Meis1^LoxP/LoxP^* mice. (**E**) Typical sinus

*Figure 1 continued on next page*

*Figure 1 continued*

arrest (SA) trace obtained in a *PLAT^CRE^::Meis1^LoxP/LoxP^* mouse and quantification of the mean number of sinus arrest per hour over a 12 hr ECG recording in *PLAT^CRE^::Meis1^LoxP/LoxP^* vs WT mice. (**F**) Mean RR interval after treadmill exercise. Typical ECG trace during the recuperation period following treadmill exercise. Data are represented as mean +/- s.e.m. n = 8 in each group; *p<0.05, ***p<0.001. See also *Figure 1—figure supplement 1* and *2* , *Figure 1—source data 1.*

The following source data and figure supplements are available for figure 1:

**Source data 1.** Morphologic and left ventricular function parameters assessed by Doppler echocardiography in WT and *PLAT^CRE^::Meis1^LoxP/LoxP^* mice.

**Figure supplement 1.** Sympatho-vagal deregulation of cardiac functions following *Meis1* inactivation in the PNS.

**Figure supplement 2.** Cardiac echography analysis revealed no difference following *Meis1* inactivation.

---

and sustained bradycardia that was lethal for 3 out of 8 mutant mice, confirming the lack of endogenous sympathetic activity to counterbalance this effect. We further performed exercise tests on the treadmill (*Figure 1F*). Whereas a significant 45% decrease in the RR interval was recorded in WT mice during the effort (p<0.01, n = 8), a non-significant and delayed 13% decrease in the RR interval was observed in *PLAT^CRE^::Meis1^LoxP/LoxP^* mice. In contrast to control WT mice, during the recuperation phase, 75% of *PLAT^CRE^::Meis1^LoxP/LoxP^* mice developed ventricular fibrillations followed by cardiac arrest that was occasionally cardio-converted as shown in *Figure 1F*.

Because *Meis1* inactivation has been linked to cardiac septum defect (*Stankunas et al., 2008*), a structure derived partly from neural crest cells (*Kirby and Waldo, 1995*), and to exclude the possible participation of such defects in the cardiac phenotype we observed, heart sections from E14.5 and E16.5 WT and *PLAT^CRE^::Meis1^LoxP/LoxP^* were analyzed. In 3/3 E14.5 and 5/5 E16.5 mutant embryos analyzed, we never observed any defect in cardiac septum closure. However, in 1/1 *Wnt1^CRE^::Meis1^LoxP/LoxP^*, a CRE driver commonly used to recombine in cardiac neural crest derivatives (*Choudhary et al., 2006*), the cardiac septum was not fully closed (*Figure 1—figure supplement 1E*).

In addition, we performed transthoracic-echocardiography measurements in the parasternal long and short axis views (*Figure 1—figure supplement 2,A and B* respectively) to characterize heart structure, contractile function and morphology. Both views did not revealed any defect in the septum of *PLAT^CRE^::Meis1^LoxP/LoxP^* mice and the ejection fraction (EF), which is used as the conventional contractile function index, was not different from WT mice (*Figure 1—figure supplement 2C* and *Figure 1—source data 1*). In addition, measuring systolic performance tracing all along endocardial end-diastolic and end-systolic borders (EF B-mode and FAC; *Figure 1—source data 1*) excluded a left ventricular abnormal regional remodeling. Finally, heart diastolic performance assessed by measuring left ventricle filling waves in standard 4 cavities view. (*Figure 1—figure supplement 2D* and E/A ratio in *Figure 1—source data 1*) did not show any difference. Thus, *PLAT^CRE^::Meis1^LoxP/LoxP^* mice present no cardiac malformations and exhibit normal hemodynamic parameters suggesting that the outflow tract is unaffected.

Altogether, these explorations revealed that *PLAT^CRE^::Meis1^LoxP/LoxP^* mice present profound alterations in the sympatho-vagal regulation of cardiac functions, affecting specifically the sympathetic component whereas the parasympathetic component is unaffected. The high occurrence of sinus arrest together with spontaneous blocks of conduction, chronotropic incompetence and episodes of ventricular fibrillation following exercise clearly point to a high increased risk for SCD in adult *PLAT^CRE^::Meis1^LoxP/LoxP^* mice dependent on altered sympatho-vagal control of heart rhythm and independent of cardiac malformations. Together with the reduced sympathetic innervation of cardiac tissue, these results led us to investigate in depth the role of Meis1 function in sympathetic neurons.

## The onset of Meis1 expression is compatible with a function in target-field innervation but not early sympathetic specification.

To decipher Meis1 function in sympathetic neurons, we first examined its expression during mouse superior cervical ganglia (SCG) formation. By in situ hybridization (ISH), we compared its mRNA expression to that of well-established early markers for sympathetic neurons and their precursors. We found that at E11.5, whereas noradrenergic differentiation has already been initiated as seen by the onset of Gata3, Phox2b and TH expressions, sympathetic neurons did not express Meis1 or Klf7 transcripts (*Figure 2A*). Meis1 mRNA expression was first faintly detected at E12.5, after the onset of noradrenergic specification in most neurons, together with Ntrk1, Klf7, and Ret, the receptor for the Gdnf family of ligands (*Figure 2B*). At E14.5, all sympathetic neurons, including neurons in the stellate and the ganglionic chain, showed a strong Meis1 mRNA expression (*Figure 2—figure supplement 1A*). The other members of the Meis family, Meis2 and Meis3, were never expressed by these neurons at any of the analyzed stages (*Figure 2—figure supplement 1A*). At E16.5, Meis1 mRNA co localized indifferently with DBH or with the presumptive marker for cholinergic sympathetic neurons Ret (*Figure 2—figure supplement 1B*). The relatively late expression of Meis1 after that of Gata3, Phox2b and TH strongly suggests that Meis1 is not involved in early sympathetic neurons specification. Immunochemistry confirmed the complete loss of Meis1 in sympathetic neurons from 14.5 and E16.5 $PLAT^{CRE}::Meis1^{LoxP/LoxP}$ mutant embryos, (*Figure 2—figure supplement 1C*). In E14.5 mutant embryos, the SCGs were normally formed and positioned, indicating that the formation, survival and migration of neural crest cells, the precursors of sympathetic neurons, were unaffected. At this stage, *Meis1* loss of function did not affect the expression of Gata3, Hand2, Phox2b and TH (*Figure 2C*). Collectively, these results demonstrate that Meis1 function is dissociated from early noradrenergic specification and early diversification of sympathetic neurons. Conversely, the simultaneous temporal expression of Meis1 with Ntrk1 and Klf7 led us to hypothesize that Meis1 is involved in later aspects of sympathetic neurons development during the period of target-field innervation and/or survival.

## *Meis1* deficient sympathetic neurons progressively die by apoptosis.

To test this hypothesis, we first investigated sympathetic neurons survival and followed the embryonic development of the SCGs in E14.5, E16.5 and E18.5 WT and $PLAT^{CRE}::Meis1^{LoxP/LoxP}$ embryos. In WT embryos, there was a 1.8 fold increase in the volume of the SCGs between E14.5 and E16.5, and a 3.3 fold increase between E14.5 and E18.5. In $PLAT^{CRE}::Meis1^{LoxP/LoxP}$ embryos, we measured a slight reduction of 25% and a dramatic reduction of 60% in the volume of the ganglia at E16.5 and E18.5 respectively (*Figure 3A and B*). We then investigated if the decreased volume of the SCGs in mutant embryos was due to neuronal apoptosis. We found an increase of Casp3$^+$ apoptotic sympathetic neurons in $PLAT^{CRE}::Meis1^{LoxP/LoxP}$ compared to WT at both E14.5 and E16.5. In E14.5 and E16.5 WT embryos, we counted 1.8 ± 0.4% and 1.3 ± 0.6% of Casp3$^+$ neurons per section respectively, whereas in $PLAT^{CRE}::Meis1^{LoxP/LoxP}$, we found 5.6 ± 1.1% and 7.6 ± 1.1% of Casp3$^+$ sympathetic neurons per section at E14.5 and E16.5 respectively (*Figure 3C*). To investigate if glial defects could contribute to the decreased ganglia volume, we conducted immunochemistry for Sox2 and ISH for Sox10, two well-described markers of peripheral glial cells and neural precursors. As expected, Sox2 never colocalized with the neuronal marker Phox2b (*Figure 3—figure supplement 1B*), and there was no change in the number of Sox2$^+$ or Sox10$^+$ glial progenitor cells at E16.5 nor in the number of Sox2$^+$ cells at E14.5 (*Figure 3—figure supplement 1A, C and D*). Casp3$^+$ cells never coexpressed Sox2 but largely colocalized with TH (*Figure 3—figure supplement 1B*). To further confirm that the neuronal loss was due to apoptosis, we generated triple $PLAT^{CRE}::Meis1^{LoxP/LoxP}::Bax^{-/-}$ mutant embryos. As expected, we found that inactivating the pro-apoptotic gene *Bax* in $PLAT^{CRE}::Meis1^{LoxP/LoxP}$ mice rescued sympathetic neurons from apoptosis and there was no significant difference in the number of Phox2b$^+$ neurons of the SCGs of newborn P0 $Bax^{-/-}$ vs $PLAT^{CRE}::Meis1^{LoxP/LoxP}::Bax^{-/-}$ mice (*Figure 3E*). Of note, Phox2b expression was not altered in *Meis1* deficient sympathetic neurons rescued from apoptosis at this stage, further confirming that Meis1 functions are independent of noradrenergic specification. Finally, because early sympathetic neurons continue to divide up to perinatal stages, and to exclude the contribution of this mechanism to the decrease in ganglia size, we evaluated their proliferation and demonstrated that the proliferative

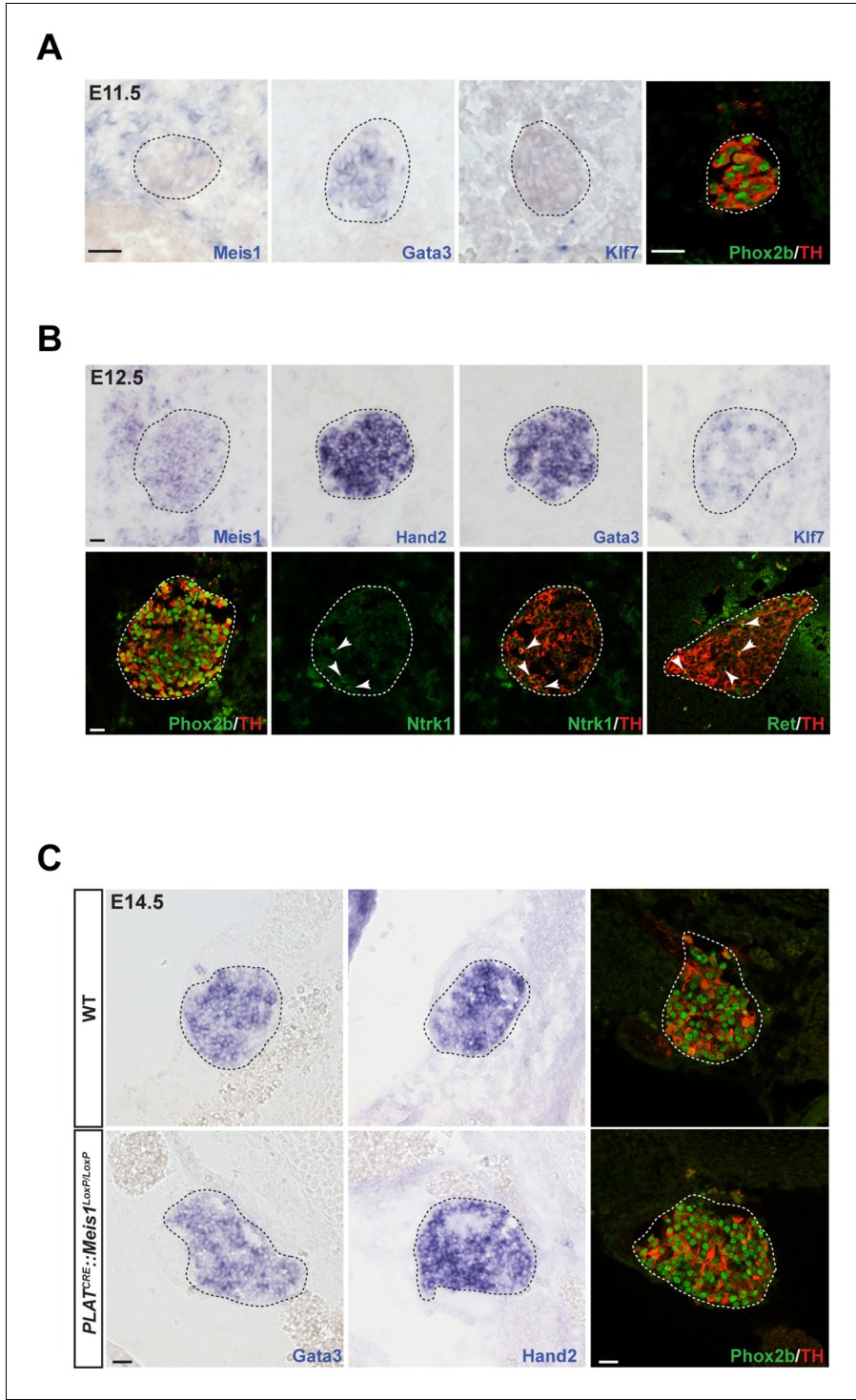

**Figure 2.** Meis1 expression in sympathetic neurons is incompatible with early sympathetic specification but coincides with target-field innervation onset. (**A**) ISH for Meis1, Gata3 and Klf7, and immunochemistry for Phox2b and TH in WT E11.5 embryonic SCGs. (**B**) ISH for Meis1, Hand2, Gata3 and Klf7 and immunochemistry for Phox2b, TH, Ntrk1 and Ret in WT E12.5 embryonic SCGs. White arrows point at Ntrk1 or Ret neurons. (**C**) ISH for Gata3 and Hand2 and immunochemistry for Phox2b and TH on WT and *PLAT^CRE^::Meis1^LoxP/LoxP^* E14.5 embryonic SCGs. White and black dotted lines delineate the SCGs. Scale bar = 20 µm. See also *Figure 2—figure supplement 1*.

The following figure supplement is available for figure 2:

*Figure 2 continued on next page*

*Figure 2 continued*

**Figure supplement 1.** *Meis1* is the only member of the *Meis* family to be expressed by embryonic sympathetic neurons and is indifferently expressed by noradrenergic and cholinergic sympathetic neurons.

rates were not affected at any of the analyzed stages in $PLAT^{CRE}::Meis1^{LoxP/LoxP}$ embryos (*Figure 3—figure supplement 2*).

In summary, the loss of SCG neurons in $PLAT^{CRE}::Meis1^{LoxP/LoxP}$ mice occurs in two phases, an early phase (E14.5 – E16.5) with a moderate increase of neuronal apoptosis and a small variation in the volume of the SCGs, a later phase (E16.5 –18.5) with a dramatic reduction in the volume of the SCGs.

## Distal target-field innervation but not proximal axonal growth is impaired in *Meis1*-inactivated sympathetic neurons.

We next investigated whether the progressive neuronal loss was a consequence of impaired proximal and/or distal axonal growth. Proximal axonal growth was unaffected as seen by the presence of sympathetic nerves exiting the SCGs in both E14.5 and E16.5 WT and $PLAT^{CRE}::Meis1^{LoxP/LoxP}$ embryos (*Figure 4A*). To test whether distal target-field innervation was affected, we investigated TH immunoreactivity in peripheral organs at E16.5 to visualize sympathetic innervation within target organs. Whereas the density of $PGP9.5^+/TH^-$ fibers representing non-sympathetic peripheral innervation was similar in WT and mutant embryos, we found a dramatic decrease in $PGP9.5^+/TH^+$ sympathetic fibers in the heart of $PLAT^{CRE}::Meis1^{LoxP/LoxP}$ compared to WT embryos (*Figure 4B and C*). Measurements revealed a $77 \pm 12\%$ reduction in $TH^+$ sympathetic innervation density of the heart in $PLAT^{CRE}::Meis1^{LoxP/LoxP}$ embryos compared to WT (*Figure 4C*). Similarly, we measured a marked reduction in $TH^+$ sympathetic innervation densities of the salivary glands, the vomeronasal organ, the tongue and the trachea of mutant embryos (*Figure 4—figure supplement 1*). In addition, preventing apoptosis of sympathetic neurons lacking *Meis1* by *Bax* mutation did not rescue the defects in sympathetic cardiac innervation at P0 (*Figure 4D*). These results demonstrated that in the absence of Meis1, sympathetic target-field innervation, but not proximal axonal growth, is compromised.

## Loss of target-field innervation signaling pathways in *Meis1*-inactivated sympathetic neurons.

According to the neurotrophic hypothesis, most sympathetic neurons depend on retrograde Ngf/Ntrk1 signaling for target-field innervation and survival. Ngf/Ntrk1 retrograde signaling is necessary for the maintenance of Ntrk1 expression and accumulation of phosphorylated CREB (pCREB) within neuronal nuclei (*Deppmann et al., 2008*; *Riccio et al., 1997*). Thus, defects in retrograde Ngf/Ntrk1 signaling would result in altered target-field innervation without proximal growth defects and a massive neuronal apoptosis concomitant to naturally occurring cell-death, a phenotype that resembles the phenotype we observed during the second phase of neuronal loss in $PLAT^{CRE}::Meis1^{LoxP/LoxP}$ embryos. To establish a possible link between Meis1 and Ngf/Ntrk1 signaling, we performed ISH and immunochemistry for Ntrk1. We found a reduction of $73 \pm 11\%$ and of $73 \pm 6\%$ in Ntrk1 ISH and immunofluorescence signal intensities respectively in the SCGs of E16.5 $PLAT^{CRE}::Meis1^{LoxP/LoxP}$ compared to WT (*Figure 5A–D*). In contrast, no change in Ntrk1 mRNA levels was observed at E14.5 (*Figure 5A*). The expression levels of the other receptors for the neurotrophins family Ntrk2, Ntrk3 and Ngfr were identical in E16.5 WT and mutant SCGs (*Figure 5—figure supplement 1A*). Together these results show that *Meis1* is necessary for the maintenance of Ntrk1 expression but not its induction. In addition, we conducted ISH for Klf7 and found a $49 \pm 3\%$ reduction in Klf7 mRNA expression in E16.5 sympathetic neurons of mutant embryos (*Figure 5E and F*). To confirm that apoptosis was not responsible for the decrease in Klf7 and Ntrk1 expressions, similar staining were conducted on sections from $PLAT^{CRE}::Meis1^{LoxP/LoxP}::Bax^{-/-}$ and $Bax^{-/-}$ P0 newborn mice, and we found a comparable decrease of $91 \pm 4\%$ in Ntrk1 expression (*Figure 5G and H*). Klf7 expression also was not maintained in $PLAT^{CRE}::Meis1^{LoxP/LoxP}::Bax^{-/-}$ newborn mice (*Figure 5E*). As a readout of retrograde Ngf/Ntrk1 signaling, we investigated pCREB immunoreactivity in sympathetic neurons nuclei. At E14.5, most of the mutant neurons failed to express pCREB (*Figure 5I*) and by E16.5,

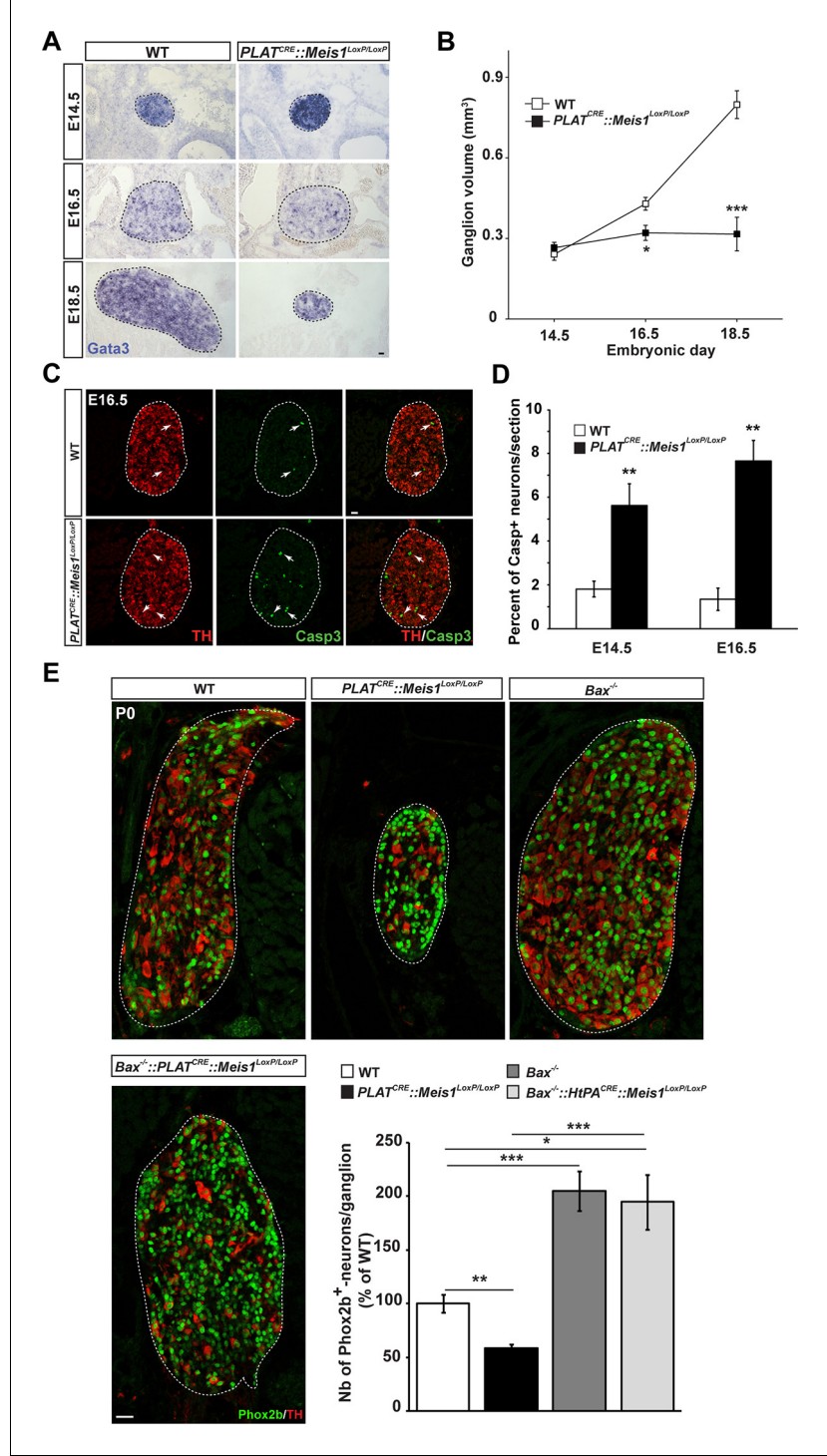

**Figure 3.** Exacerbated neuronal apoptosis in *Meis1* mutant sympathetic neurons. (**A**) ISH for Gata3 on SCGs of *PLAT*[CRE]*::Meis1*[LoxP/LoxP] and WT embryos at E14.5, 16.5 and E18.5. (**B**) Quantification of the volume of the SCGs in WT *vs PLAT*[CRE]*::Meis1*[LoxP/LoxP] mice at E14.5, E16.5 and E18.5. (**C**) Representative images of the SCGs visualized by immunochemistry for activated Caspase-3 (Casp3) and TH at E16.5 in WT and *PLAT*[CRE]*::Meis1*[LoxP/LoxP] mice. Arrows point at Casp3[+]/TH[+] neurons. (**D**) Quantification of the average number of Casp3[+]/TH[+] neurons per section in the SCGs of *PLAT*[CRE]*::Meis1*[LoxP/LoxP] and WT embryos at E14.5 and E16.5. (**E**) Immunochemistry for Phox2b and TH on SCGs of P0 WT, *Bax*[-/-], *PLAT*[CRE]*::Meis1*[LoxP/LoxP] and *PLAT*[CRE]*::Meis1*[LoxP/LoxP]*::Bax*[-/-] newborn pups, and quantification of the number of Phox2b[+] neurons per SCG in P0 WT, *Bax*[-/-], *PLAT*[CRE]*::Meis1*[LoxP/LoxP] and *Figure 3 continued on next page*

*Figure 3 continued*

*Bax*$^{-/-}$*::PLAT*$^{CRE}$*::Meis1*$^{LoxP/LoxP}$ mice. Data are represented as mean +/- s.e.m. n = 3; *p≤0.05; **p≤0.01; ***p≤0.005. Scale bar = 20 μm. See also *Figure 3—figure supplement 1* and *2*.

The following figure supplements are available for figure 3:

**Figure supplement 1.** Survival of glial progenitor cells is not affected following *Meis1* inactivation.

**Figure supplement 2.** *Meis1* inactivation does not affect early proliferation of sympathetic neurons.

pCREB signal intensity was largely reduced in all the nuclei of mutant neurons compared to WT (*Figure 5J*), indicating that downstream signaling of Ngf/Ntrk1 retrograde signaling is perturbed following *Meis1* inactivation. A possible explanation of these defects could be that Meis1 itself is a downstream effector of Ngf/Ntrk1 retrograde signaling. To test this hypothesis, we performed immunochemistry for Meis1 on SCGs from E17.5 Ngf$^{-/-}$ embryos. We found that Meis1 expression was unchanged when *Ngf* is inactivated (*Figure 5—figure supplement 1B*), indicating that it is not a downstream effector of Ngf/Ntrk1 signaling.

All together, these results demonstrate that Ngf/Ntrk1 retrograde signaling is altered following *Meis1* inactivation. The decreased expression of Ntrk1 at E16.5 but not E14.5 suggests that during the early and late phases of cell loss in *Meis1* mutants, different signaling pathways are affected. In addition, whereas the rescue of *Meis1* inactivated neurons in a *Bax* null-context support the involvement of Ngf/Ntrk1 retrograde signaling in the late phase of neuronal apoptosis, the loss of innervation in the trachea clearly indicates that target-derived Ngf is not responsible alone, and that a more general mechanism is affected.

## Impaired early endosome formation and traffic in *Meis1* inactivated neurons.

One hypothesis is that Meis1 controls the expression of effector genes that are functionally related to retrograde vesicular transport. Because Meis1 target genes in the nervous system are largely unidentified and to identify such effector genes, we performed a ChIP-seq analysis on SCGs of E16.5 WT embryos. High throughput sequencing identified putative regulatory regions near 309 potential target genes (*Figure 6—source data 1*). Bibliographical analyses revealed that about 34% of these genes are involved in target-field innervation and synapse function and/or architecture or both (*Figure 6—figure supplement 1A*). We also found that many of the Meis1 potential target genes have been documented to participate in heart functions or pathologies (*Figure 6—figure supplement 1B* and C and *Figure 6—source data 2*). Strikingly, a large number of the identified genes (65/309) were involved in vesicles formation, traffic and fusion or the regulation of exocytosis and/or endocytosis (*Figure 6—source data 3*). To see if such genes were transcriptionally regulated by Meis1, we selected a number of them for deeper analysis by ISH in WT and *PLAT*$^{CRE}$*::Meis1*$^{LoxP/LoxP}$ embryos. We found that the expression of synaptotagmin1 (Syt1) mRNA was totally lost in E14.5 and E16.5 *Meis1* mutants embryos (*Figure 6A*). Similarly, the expressions of Lunapark (Lnp), Dystonin/Bpag1 (Dst), Neurobeachin (Nbea) and Adam19 were strongly reduced in E16.5 mutant embryos compared to WT (*Figure 6—figure supplement 1D*). Because endocytosis and retrograde transport-mediated survival and distal axonal growth are hallmarks of embryonic peripheral neurons and because Syt1 and Dst physically interact with clathrin, we investigated whether such processes were altered following *Meis1* loss of function. Visualizing clathrin coated vesicles by immunochemistry in E16.5 *PLAT*$^{CRE}$*::Meis1*$^{LoxP/LoxP}$ embryos revealed a dramatic decrease in clathrin-positive structures in the sympathetic axons innervating the heart, as well as in the cell bodies of sympathetic neurons (*Figure 6B,C*), suggesting perturbation in the traffic of endosomal cargoes.

To better discriminate vesicular staining in sympathetic axons, we cultured SCGs explants from WT and *PLAT*$^{CRE}$*::Meis1*$^{LoxP/LoxP}$ E16.5 embryos. In vitro, axonal extension was not statistically different in sympathetic neurons from *PLAT*$^{CRE}$*::Meis1*$^{LoxP/LoxP}$ and WT embryos (*Figure 7A*), confirming that proximal axonal growth is not affected by *Meis1* inactivation. In these cultures, immunochemical analysis demonstrated a strong reduction in the numbers of clathrin$^{+}$, Synaptophysin-1$^{+}$ (Syp1) and Rab5$^{+}$ cargoes in *Meis1*-inactivated sympathetic axons compared to WT (*Figure 6D,E*). To further

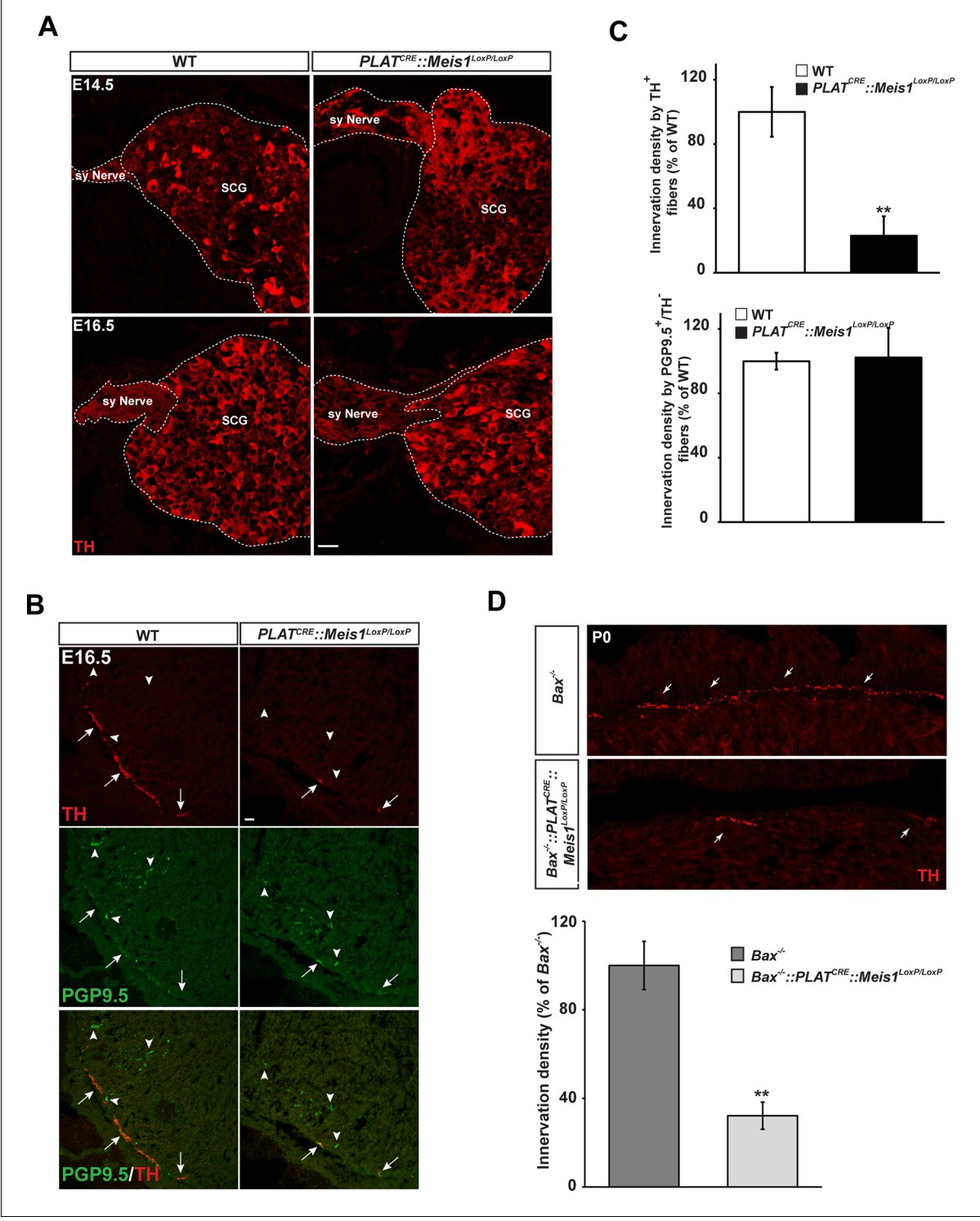

**Figure 4.** Lack of terminal sympathetic target-field innervation in *Meis1* mutant embryos. (**A**) Immunochemistry for TH on SCGs from E14.5 and E16.5 WT and *PLAT^CRE^::Meis1^LoxP/LoxP^* embryos showing that proximal axonal growth of sympathetic neurons is not affected. Sy Nerve = sympathetic nerve. (**B**) Visualization of TH+ sympathetic fibers and others PGP9.5+/TH- peripheral neurons projections in the heart of *PLAT^CRE^::Meis1^LoxP/LoxP^* and WT embryos at E16.5. Arrows point at TH+ sympathetic fibers and arrowheads point at PGP9.5+/TH- nerves. (**C**) Quantification of the density of TH+/PGP9.5+ sympathetic and TH-/PGP9.5+ sensory fibers on heart sections of *PLAT^CRE^::Meis1^LoxP/LoxP^* and WT embryos at E16.5. (**D**) Representative images and quantification of TH+ sympathetic fibers in the heart of *Bax^-/-^* and *PLAT^CRE^::Meis1^LoxP/LoxP^::Bax^-/-^* newborn pups. Data are represented as mean +/- s.e.m. n = 3; **p≤0.01. Scale bar = 20 μm. See also *Figure 4—figure supplement 1*.

*Figure 4 continued on next page*

*Figure 4 continued*

The following figure supplement is available for figure 4:

**Figure supplement 1.** Sympathetic target-field innervation of peripheral organs is compromised following *Meis1* loss of function.

investigate the traffic of endosomal cargoes, time-lapse imaging was performed on SCGs cultured from WT and $PLAT^{CRE}::Meis1^{LoxP/LoxP}$ E16.5 embryos. Overexpressing a Rab5-GFP construct in mutant neurons did not rescue the number of Rab5$^+$ endosomes, suggesting that the loss of Rab5 immunoreactivity is due to a loss of endosomal structures rather than a loss of Rab5 protein expression (*Figure 6D*). Accordingly, analysis of E16.5 sympathetic nerves by transmission electron microscopy (TEM) showed that the number and the surface area occupied by endocytic structures per axon section were reduced in mutant nerves (*Figure 6—figure supplement 2*). Time-lapse analysis on SCGs explants indicates that the percentages of motile, including moving and oscillating, Rab5-GFP$^+$ endosomes were dramatically reduced in mutant neurons, whereas the percentage of pausing Rab5$^+$ cargoes were accordingly increased (*Figure 7B and C*; *Videos 1–4*). In addition, the velocity of both anterograde and retrograde Rab5-GFP$^+$ endosomes was reduced in mutant neurons (*Figure 7D,E*). Altogether, these results demonstrated that in the sympathetic nervous system, part of the transcriptional program mediated by Meis1 is necessary for Rab5$^+$ endocytic structures formation and traffic.

## Discussion

In this work, we demonstrated that genetic inactivation of *Meis1* in rodent results in an imbalanced sympatho-vagal regulation of heart functions leading to conduction defects, chronotropic incompetence and premature SCD. In addition, we demonstrated that Meis1 acts independently of the well-studied network of transcription factors governing early specification of sympathetic neurons to orchestrate later phases of neuronal differentiation. Its inactivation results in defects of distal innervation and progressive neuronal apoptosis. Identification of Meis1 effector genes revealed a large number of genes involved in endosomal and clathrin-coated cargoes formation and fusion, as well as cytoskeletal proteins participating in their anchoring and transport. Accordingly, loss of *Meis1* function impairs Rab5$^+$ endosomes formation and trafficking.

The multilevel origin in the trigger, the persistence and the termination of a variety of inherited arrhythmias such as long QT syndrome, Brugada syndrome, progressive cardiac conduction block, sinus node dysfunction, catecholaminergic polymorphic ventricular tachycardia or sudden infant death syndrome is well established (*Keating and Sanguinetti, 2001*), and include an imbalanced autonomic regulation. For instance, studies in mice and humans have revealed the implication of Sema3a-mediated sympathetic axonal guidance and patterning in the development of ventricular tachycardia and SCD (*Ieda et al., 2007*; *Nakano et al., 2013*; *Fukuda et al., 2015*). Here, we demonstrated that *Meis1* inactivation results at adult stages in a persistent and spontaneous bradycardia, blocks of conduction and severe sinus arrests independently of any cardiac malformation. The situation irreversibly deteriorated in conditions requiring sympathetic activation to restore cardiac homeostasis such as drug administration or the recuperation period following physical activity, thus demonstrating the complete chronotropic incompetence of these mice. In humans, such chronic heart rate abnormalities are associated with the lethal dysrhythmias observed in athletes (*Link et al., 2002*). Similarly, for infant SCD in which the autonomic regulation is compromised, severe hypoxia or anoxia during sleep lead to lethal bradycardia because autoresuscitation, also known as the Lazarus phenomenon, cannot be initiated to restore heart rate, blood pressure and breathing (*Stramba-Badiale et al., 1992*; *Yun and Lee, 2004*). Thus, a failure in sympathetic activation during this phase leads to unexpected and sudden death. Such an increased risk of SCD following exercise and during sleep has also been reported in dogs with inherited spontaneous cardiac arrhythmia (*Dae et al., 1997*). Similar to our results, in these dogs, the sympathetic innervation is defective. A previous study on E14.5 $Meis1^{-/-}$ embryos has revealed ventricular septal defect and an overriding aorta in both right and left ventricles (*Stankunas et al., 2008*). In our study, $PLAT^{CRE}::Meis1^{LoxP/LoxP}$ embryos did not present such gross structural defects as evidenced by histologic and echocardiographic

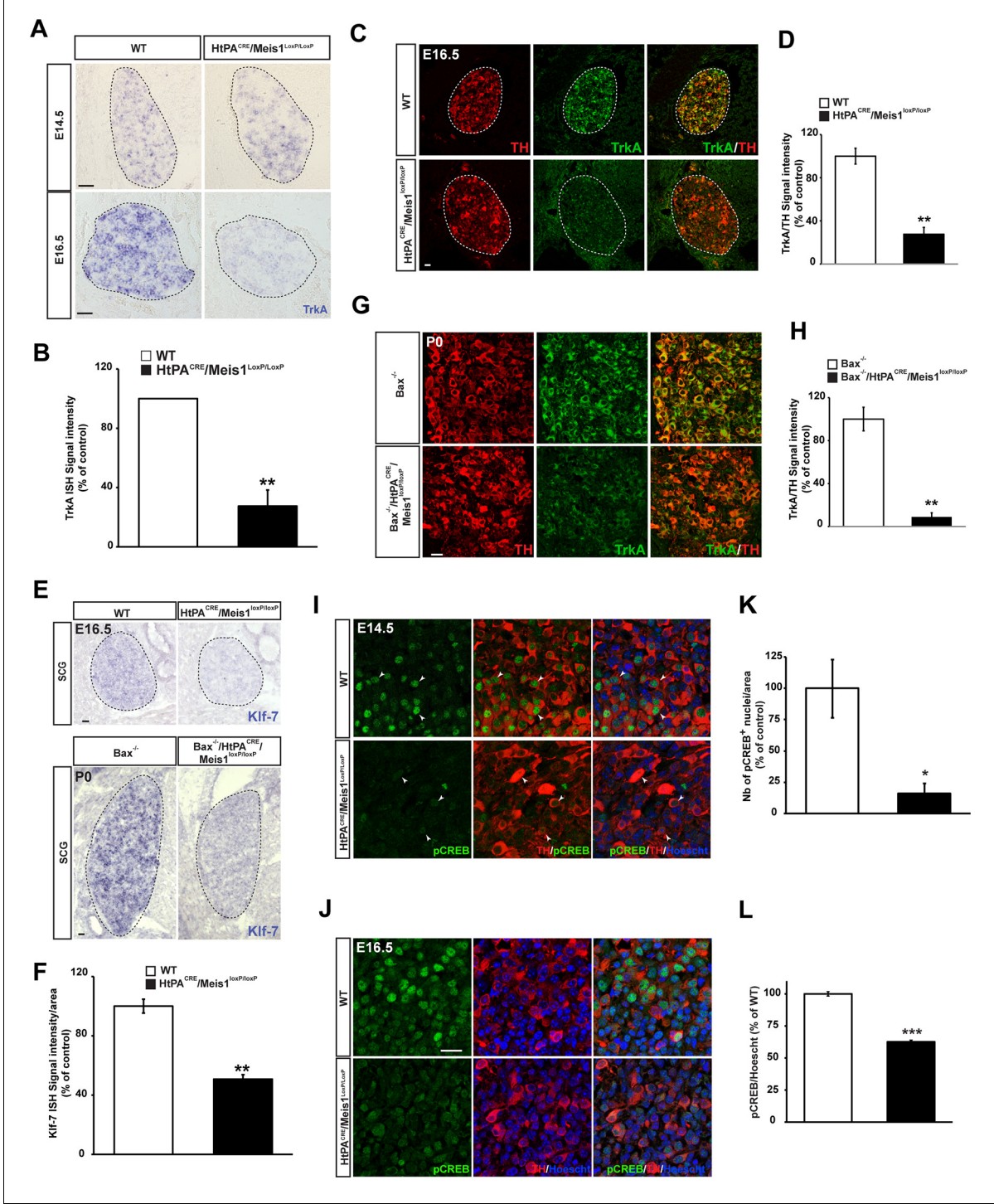

**Figure 5.** Target-field innervation signaling pathways in *Meis1*-inactivated sympathetic neurons. (**A**) ISH for the neurotrophins high affinity receptor Ntrk1 in E14.5 and E16.5 WT and *PLAT*<sup>CRE</sup>*::Meis1*<sup>LoxP/LoxP</sup> SCGs. (**B**) Quantification of Ntrk1 ISH signal intensity in E16.5 WT and *PLAT*<sup>CRE</sup>*::Meis1*<sup>LoxP/LoxP</sup> SCGs. (**C**) Immunochemistry for TH and Ntrk1 on E16.5 WT and *PLAT*<sup>CRE</sup>*::Meis1*<sup>LoxP/LoxP</sup> embryos. White dotted lines encircle the SCGs. (**D**) Quantification of the intensity of immuno-fluorescence for Ntrk1 in the SCGs of E16.5 WT and *PLAT*<sup>CRE</sup>*::Meis1*<sup>LoxP/LoxP</sup> embryos. (**E**) Representative images of ISH for Klf7 on SCGs of E16.5 WT and *PLAT*<sup>CRE</sup>*::Meis1*<sup>LoxP/LoxP</sup> embryos and on SCGs of P0 *Bax*<sup>-/-</sup> and *PLAT*<sup>CRE</sup>*::Bax*<sup>-/-</sup>*::Meis1*<sup>LoxP/LoxP</sup> mice. (**F**) Quantification of Klf7 ISH signal intensity in SCGs of E16.5 WT and *PLAT*<sup>CRE</sup>*::Meis1*<sup>LoxP/LoxP</sup> embryos. (**G**) Immunochemistry for TH and Ntrk1 on P0 *Bax*<sup>-/-</sup> and *PLAT*<sup>CRE</sup>*::Meis1*<sup>LoxP/LoxP</sup>*::Bax*<sup>-/-</sup> mice. (**H**) Quantification of the intensity of Ntrk1 immuno-fluorescence in the SCGs of P0 *Bax*<sup>-/-</sup> and *PLAT*<sup>CRE</sup>*:: Meis1*<sup>LoxP/LoxP</sup>*::Bax*<sup>-/-</sup> mice. (**I**) Immunochemistry for pCREB and TH on E14.5 SCGs from WT and *PLAT*<sup>CRE</sup>*::Meis1*<sup>LoxP/LoxP</sup> embryos and quantification of the number of pCREB<sup>+</sup> nuclei. (**J**) Immunochemistry for pCREB and TH on E16.5 SCGs from WT and *PLAT*<sup>CRE</sup>*::Meis1*<sup>LoxP/LoxP</sup> embryos and quantification
*Figure 5 continued on next page*

*Figure 5 continued*

of pCREB signal intensity in neuronal nuclei. Data are represented as mean +/- s.e.m. n = 3; *p≤0.05; **p≤0.01. Scale bar = 20 μm. See also *Figure 5— figure supplement 1* .

The following figure supplement is available for figure 5:

**Figure supplement 1.** *Meis1* loss of function does not interfere with the expression of other neurotrophins receptors than Ntrk1 and its expression is not dependent on Ngf/Ntrk1 signaling.

measure of several parameters. Observation of the septum in the long and short axis and functional evaluation of the outflow did not reveal any differences between WT and *PLAT^CRE^::Meis1^LoxP/LoxP^* mice. In particular, a defect in the septum of adult *PLAT^CRE^::Meis1^LoxP/LoxP^* mice would have resulted in dissimilar measures of systolic and diastolic performances due to hemodynamic and rigidity variations. Nevertheless, a septal defect was present when the *Meis1^LoxP/LoxP^* strain was crossed with another strain well-known to recombine in neural crest derivatives, the *Wnt1^CRE^* strain, confirming the previously reported function of *Meis1* in cardiac neural crest (*Stankunas et al., 2008*). This difference could be due to an earlier expression of the CRE recombinase when driven by the *Wnt1* promoter compared to the *PLAT*. Another explanation could be a lower recombination efficiency of the *PLAT^CRE^* strain specifically in the cardiac neural crest lineage. We conclude that gross morphological alterations similar to those previously reported (*Stankunas et al., 2008*) do not contribute to the phenotype reported here.

Our results further complement the study by Mahmoud *et al*. demonstrating *Meis1* function in the regulation of neonatal cardiomyocytes mitosis especially following myocardial infarction (*Mahmoud et al., 2013*). However, cardiac functions were not affected following specific *Meis1* inactivation or overexpression in cardiomyocytes (*Mahmoud et al., 2013*). Finally, because the *Meis* family participates in cardiogenesis (*Stankunas et al., 2008*; *Mahmoud et al., 2013*) and has been associated with SCD by independent GWA studies (*Pfeufer et al., 2010*; *Smith et al., 2011*), it is tempting to speculate that *Meis1* mutations in humans would result in heart dysfunctions due to cardiac malformations and/or regeneration, sympatho-vagal deregulation of cardiac rhythmicity, or a combination of dysfunctions affecting both compartments.

The onset of Meis1 expression after the early noradrenergic specification transcription factors and noradrenergic markers together with the unaltered expression of Gata3, Phox2b, Hand2 and TH at E14.5 following *Meis1* inactivation demonstrated that Meis1 acts independently of the pathways regulating early sympathetic neurons specification. Accordingly, Phox2b expression remained unchanged at later embryonic stages and at P0 in a double *Meis1::Bax* null context. These findings were confirmed by the normal formation of the sympathetic ganglia up to E14.5 in *Meis1* mutants, whereas in *Ascl1, Phox2b, Hand2* or *Gata3* null mutants, early sympathetic neurons lose TH and DBH expression, fail to proliferate and to coalesce in a ganglion and precociously die upon arrival in the sympathetic anlage (*Rohrer, 2011*).

In Drosophila, inactivation of *Homothorax (Hth)*, the ortholog of *Meis1*, leads to neuronal and dendritic patterning defect in the PNS (*Kurant et al., 1998*; *Ando et al., 2011*; *Baek et al., 2013*). Hth is also required for survival of a subset of postmitotic motoneurons and rescuing them from apoptosis demonstrated that Hth shapes axons and dendrites by regulating a set of target genes independently of Hox functions (*Baek et al., 2013*). However, the precise mechanism and the transcriptional targets by which Hth mediates these effects have not been elucidated. We propose that the regulation of the expression of genes involved in early endosomes formation and transport by Meis1 is a means of coordinating target-field innervation and survival. Accordingly, our identification of Meis1 effector genes highlighted numerous key genes for vesicles formation, fusion and transport, thus providing support for a coordinated action of Meis1 in retrograde dependent target-field innervation and neuronal survival.

The transcriptional program we identified by ChIP-seq is in broad agreement with ChIP-seq performed by others on other tissues (*Wilson et al., 2010*; *Penkov et al., 2013*). In a ChIP-seq performed on E11.5 trunk mouse embryos, ontology analysis of potential Meis1 target genes identified among the most enriched Meis1 bound sequences, genes involved in nervous system development, axonal guidance and synapse organization (*Penkov et al., 2013*). More surprisingly, comparison of

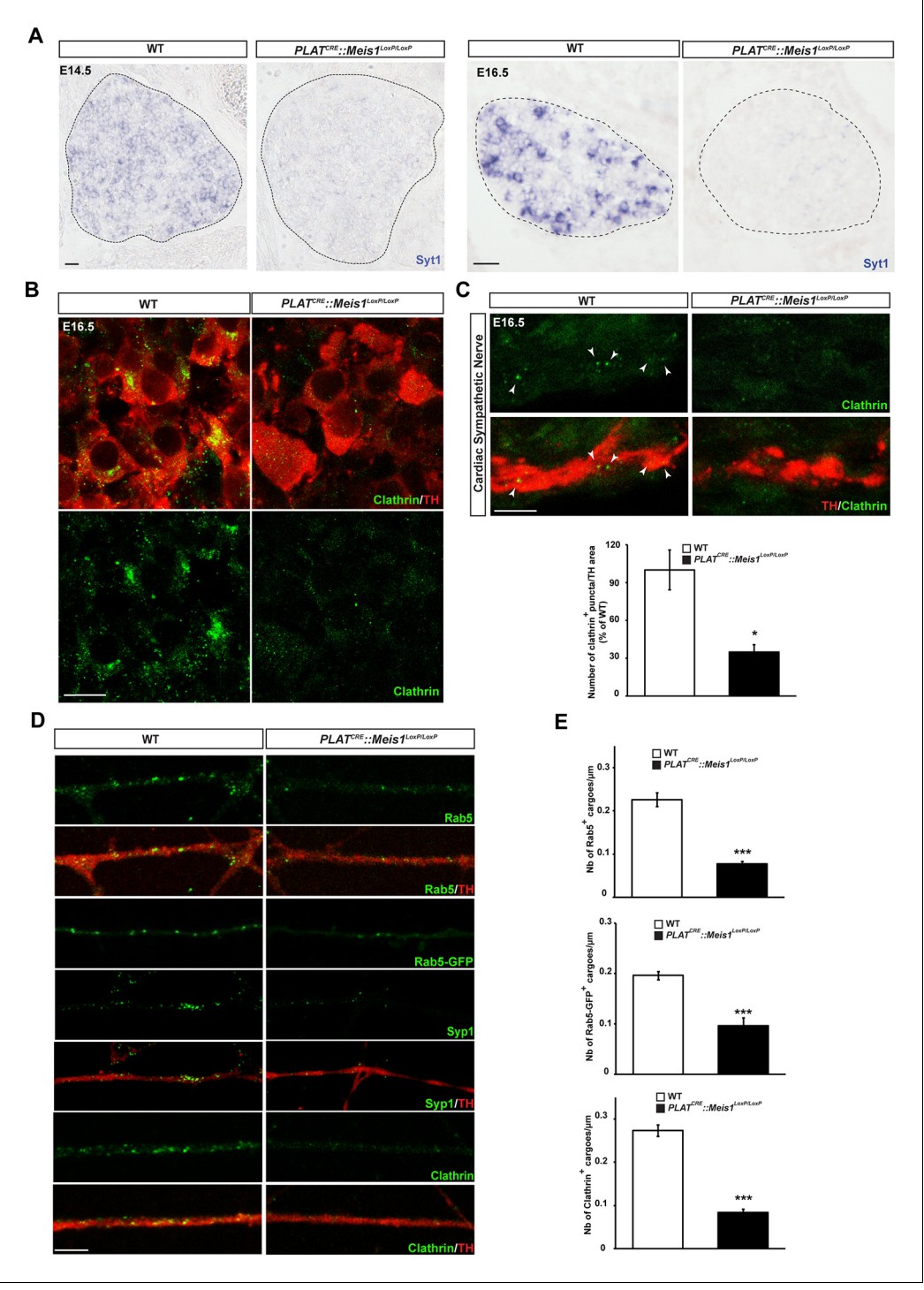

**Figure 6.** Meis1 target genes encode proteins necessary for early endosomes formation. (**A**) ISH for Syt1 on SCGs of E14.5 and E16.5 WT and *PLAT^CRE^::Meis1^LoxP/LoxP^* embryos. (**B**) Immunochemistry for clathrin and TH on SCGs of E16.5 WT and *PLAT^CRE^::Meis1^LoxP/LoxP^* embryos. (**C**) Representative images and quantification of immunochemistry for clathrin and TH on the sympathetic nerves within the heart of E16.5 WT and *PLAT^CRE^::Meis1^LoxP/LoxP^* embryos. White arrowheads point at clathrin-coated pits and cargoes. (**D**) Immunochemistry for clathrin, TH, Syp1 and Rab5 and overexpression of a Rab5-GFP construct in cultured explants of SCGs from WT and *PLAT^CRE^::Meis1^LoxP/LoxP^* embryos. (**E**) Quantification of the numbers of Rab5, Rab5-GFP and clathrin+ cargoes in cultured explants of SCGs. Dotted lines encircle the SCGs. Scale bar = 20 μm in A, 10 μm in B and 5 μm in C and D. Data are represented as *Figure 6 continued on next page*

*Figure 6 continued*

mean +/- s.e.m; n = 3; *p≤0.05; *** p≤0.005. See also *Figure 6—figure supplement 1* and *2*, and *Figure 6—source data 1–3* .

The following source data and figure supplements are available for figure 6:

**Source data 1.** Meis1 target genes in embryonic sympathetic neurons.

**Source data 2.** Bibliographical-based classification of Meis1 potential target genes.

**Source data 3.** Bibliographical classification of Meis1 potential target genes according to their function in endocytosis, exocytosis and vesicles transport.

**Figure supplement 1.** Bibliographical classification of Meis1 potential target genes.

**Figure supplement 2.** Analysis of endocytic ultrastructures in sympathetic mutant fibers.

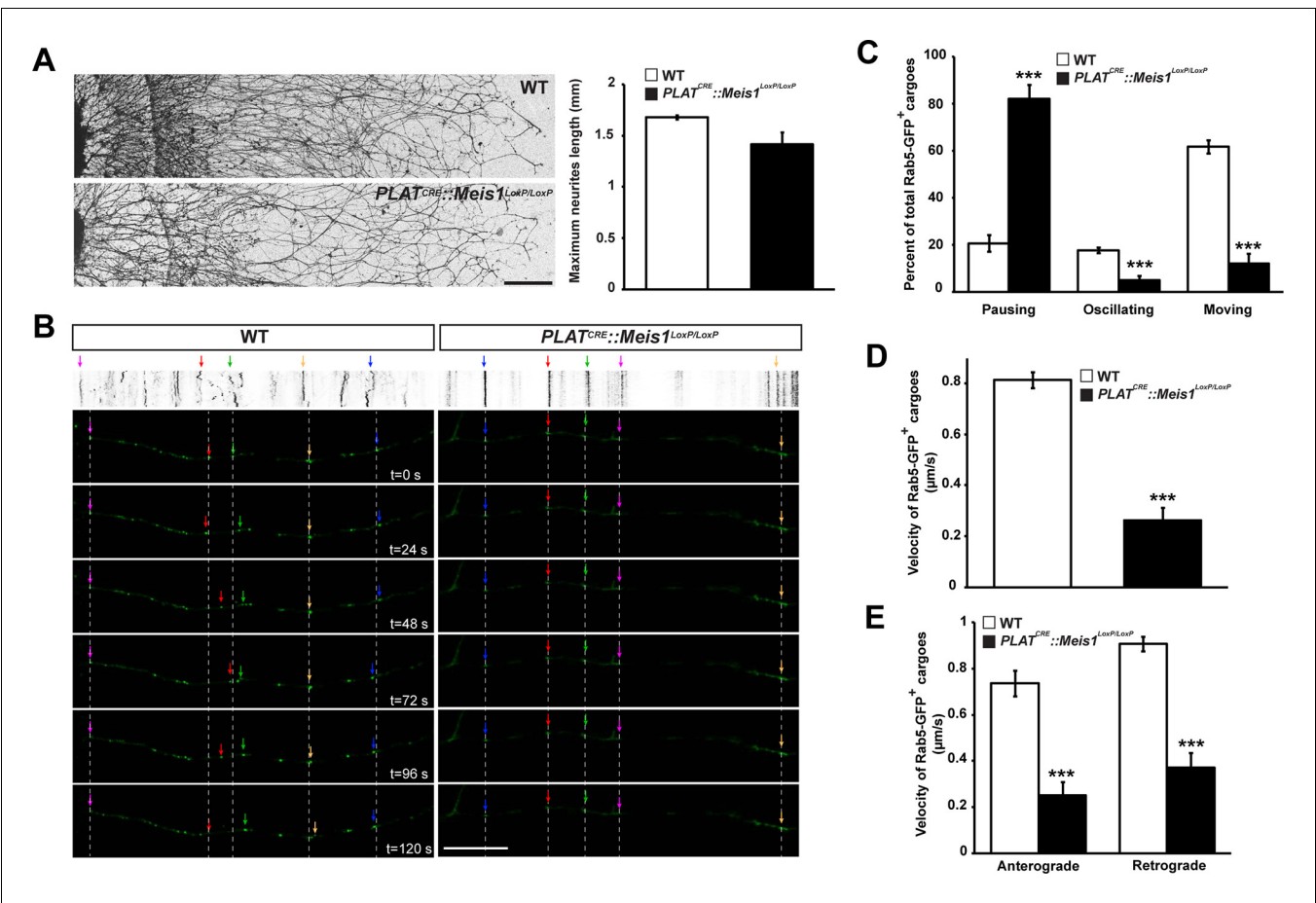

**Figure 7.** *Meis1* is necessary for early endosomes trafficking. (**A**) Camera lucida of TH-immunochemistry and quantification of neurites length on SCGs explants form WT and *PLAT^CRE^::Meis1^LoxP/LoxP^* embryos. (**B**) Individual frames and corresponding kymograph of a 2 min time lapse video showing the movement of Rab5-GFP⁺ endosomes in WT (*Video 1*) and *PLAT^CRE^::Meis1^LoxP/LoxP^* (*Video 3*) cultured SCGs explants. (**C**) Analysis of the percentage of the number of Rab5-GFP⁺ endosomes that are pausing, moving or oscillating in WT and *PLAT^CRE^::Meis1^LoxP/LoxP^* cultured SCGs explants. (**D**) Measure of the velocity of Rab5-GFP⁺ endosomes that are moving in WT and mutant conditions. (**E**) Measure of the velocity of Rab5-GFP⁺ endosomes that are moving retrogradely and anterogradely in WT and *PLAT^CRE^::Meis1^LoxP/LoxP^* cultured SCGs explants. Data are represented as mean +/- s.e.m; n = 3; ***p≤0.005. Scale bar = 200 μm in **A**, 20 μm in **B**. See also *Videos 1–4*.

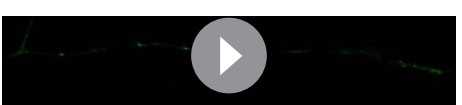

**Video 1.** Traffic of Rab5-GFP⁺ endosomes in WT neurons. Representative videos showing 2 min acquisition of Rab5-GFP⁺ endosomes trafficking in the axons of cultured WT sympathetic neurons. See also *Figure 7*.

our ChIP-seq results with results of a ChIP-seq conducted on a hematopoietic progenitor cell line revealed 37% of common potential targets between these two tissues (*Wilson et al., 2010*). Strikingly, these shared genes represent 41, 56, 75 and 50% of the genes we classified as involved in synaptic machinery, in clathrin/dynein complexes, in microtubule-associated axonal growth and in the regulation of endocytosis respectively (*Figure 6—source data 3*). From our ChIP-seq analysis, we independently confirmed that the expression of Syt1, Lnp, Adam19, Nbea and Dst mRNAs is modified. Although endosomal structures present a high diversity in their shapes and molecular compositions depending on the process they participate in, they also share a number of proteins assuming similar roles. Syt1 is an example of protein present in most, if not all, vesicular types where it acts as calcium sensor for exocytosis and endocytosis (*Kraszewski et al., 1995*; *Takei and Haucke, 2001*; *Rizo and Rosenmund, 2008*). Syt1 serves, just like synaptophysin, as a high affinity receptor at the plasma membrane of nerves-endings for clathrin AP-2, therefore participating in clathrin coated-pit nucleation and endocytosis (*Zhang et al., 1994*; *Haucke and De Camilli, 1999*). Syt1 also promotes axonal filopodia formation and branching in fibroblasts and embryonic chick forebrain neurons in culture (*Feany and Buckley, 1993*; *Greif et al., 2013*). We therefore conclude that at least part of the transcriptional program governed by Meis1 influences endocytosis and retrograde transport. In line with this, we demonstrated in E16.5 embryos, a time when retrograde transport is critical for neuronal survival and target-field innervation, that clathrin immunoreactivity is virtually absent from *Meis1* inactivated sympathetic neurons. In sympathetic neurons growing from cultured SCGs explants, the numbers of Rab5⁺, Syp1⁺ and clathrin⁺ cargoes were also reduced, and overexpressing Rab5 was unable to rescue the number of Rab5⁺ endosomes. In addition, time-lapse analysis demonstrated profound alterations in the traffic and velocity of the remaining Rab5⁺ endosomes, and TEM analysis showed that within sympathetic axons from mutant embryos the number as well as the surface area occupied by endocytic figures is reduced. We thus conclude that the transcriptional program initiated by Meis1 interferes with the retrograde transport of trophic factors as was hypothesized for Hth in Drosophila motor neurons (*Baek et al., 2013*).

Because many organelles and various kind of vesicles use similar transport machinery, the loss of expression of universal components such as Syt1, Syp1, Clathrin and Rab5 raises the possibility that other mechanisms than Ngf/Ntrk1 retrograde transport are affected. Indeed, our result are consistent with a function of Meis1 interfering with the retrograde transport of several trophic factors.

In our study, we detected an increased apoptosis of *Meis1* inactivated sympathetic neurons at E14.5 that did not result in a significant variation in the volume of the SCGs, and at E16.5 concomitantly with a significant reduction in the volume of the mutant SCGs. By E18.5, the volume of the mutant SCGs was massively reduced. Whereas the increased apoptosis at E16.5 and the further dramatic decrease of SCG volume at E18.5 coincides with the period of retrograde-signal mediating target-field innervation and survival, early apoptosis at E14.5 does not. Similarly, because in *Ngf* knockout mice, the defects in distal sympathetic innervation does not affect the trachea (*Glebova and Ginty, 2004*), we tentatively conclude that *Meis1* inactivation not only impairs Ngf/Ntrk1 retrograde signaling but also other retrograde signals important for survival. Among potential retrograde signaling pathways influencing sympathetic neurons survival, members of the Gdnf family of ligands and their receptors play important functions. They participate in various aspects of embryonic sympathetic development including precursors migration, proliferation, axonal growth and neurons survival (*Ernsberger, 2008*). Early sympathetic neurons (E12) express Ret, GFRα1, 2 and 3 (*Nishino et al.,*

**Video 2.** Impaired traffic of Rab5-GFP⁺ endosomes in *Meis1* inactivated neurons. Representative videos showing 2 min acquisition of Rab5-GFP⁺ endosomes trafficking in the axons of cultured *PLAT^CRE^::Meis1^LoxP/LoxP^* sympathetic neurons. See also *Figure 7*.

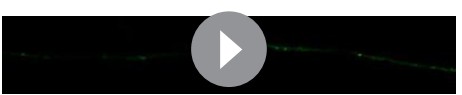

**Video 3.** Traffic of Rab5-GFP[+] endosomes in WT neurons. Representative videos showing 2 min acquisition of Rab5-GFP[+] endosomes trafficking in the axons of cultured WT sympathetic neurons. See also *Figure 7*.

1999). Artn serves as an intermediate target derived trophic factor for sympathetic neurons (*Honma et al., 2002*). In vitro, it influences the proliferation of neuroblasts from E12 to E14 and exerts a transient survival effect on newly generated neurons during this period (*Andres et al., 2001*). In agreement, in *GFRα3* KO the proliferation of neuroblasts is affected (*Andres et al., 2001*), but inactivation of *Artn* or *GRFα3* does not induce apoptosis at E14.5 (*Honma et al., 2002*). In *Ret* mutants, SCGs are mislocated and axonal growth defects are found starting E10.5 to E13 (*Enomoto et al., 2001*). However, no increase in Casp3 staining was found before E16.5 and apoptosis is enhanced only perinataly (*Enomoto et al., 2001*). In *Gdnf* mutants, there is a 35% reduction in the number of SCG neurons at P0 but earlier embryonic stages have not been investigated (*Moore et al., 1996*). However, in vitro, Gdnf promotes the survival of 30% of E13.5 rat sympathetic neurons (*Heermann et al., 2013*). Whether Gdnf can be retrogradely transported by sympathetic neurons remains controversial. Studies in neonatal and adult sympathetic indicate that Gdnf is not a retrograde survival factor (*Leitner et al., 1999*; *Tsui and Pierchala, 2010*), whereas others reported in rat embryonic neurons that application of Gdnf to the distal compartment induces axonal growth and survival via a retrograde signaling involving GFRα1 (*Coulpier and Ibáñez, 2004*). The expression of Gdnf and Artn in various intermediate and/or final target organs of sympathetic neurons during development (*Buj-Bello et al., 1995*; *Ebendal et al., 1995*; *Honma et al., 2002*; *Miwa et al., 2010*) suggests that these factors may be retrogradely transported at least during the embryonic period. It is also possible that other yet unidentified factors use retrograde transport to mediate target-field innervation and survival. We conclude that the dramatic decrease in the number of sympathetic neurons we observed at E18.5 and the general defect in target organs innervation are likely to result from impaired retrograde signaling of more than one trophic factor. Among them, because of its well-established function in target-field innervation and survival, we focused our study on retrograde Ngf/Ntrk1 downstream signaling as a proof of impaired retrograde signaling. In addition, the molecular relation between Rab5[+] endosomes and neurotrophins retrograde signaling including retrograde Ntrk1/Ngf signaling has been extensively documented (*Deinhardt et al., 2006,*, *2007*; *Varsano et al., 2006*; *Cui et al., 2007*; *Liu et al., 2007*; *Valdez et al., 2007*; *Philippidou et al., 2011*). Indeed, downstream readout of Ngf/Ntrk1 retrograde signaling *i.e.* the nuclear expression of phosphorylated CREB (*Riccio et al., 1997*) and the level of Ntrk1 expression confirmed an impairment of Ngf/Ntrk1 retrograde signaling. This is supported by the decreased number of pCREB[+] neurons at E14.5 when the first small subset of Ngf responsive sympathetic neurons are observed (*Coughlin and Collins, 1985*; *Korsching and Thoenen, 1988*), and the decreased nuclear expression of pCREB at E16.5 when survival of most sympathetic neurons depends on retrograde Ngf/Ntrk1 signaling (*Glebova and Ginty, 2004*; *Kuruvilla et al., 2004*). Similarly, because in sympathetic neurons the maintenance of Ntrk1 expression depends on retrogradely transported Ngf and because Klf7 is regulated by Ngf (*Deppmann et al., 2008*), the down regulation of Ntrk1, and possibly Klf7 we report in *Meis1* mutant is likely to be a consequence of impaired retrograde signaling of this neurotrophin. However, although the participation of Ngf/Ntrk1 retrograde signaling in the neuronal loss observed after E16.5 is supported by the loss of Ntrk1 expression and by the rescue of *Meis1* inactivated sympathetic neurons in a null *Bax* context at P0, we have no direct evidence of an Ngf dependent apoptosis. In line with this, although it was initially proposed that the survival effect of retrograde Ngf/Ntrk1 signaling was mediated through CREB activation (*Riccio et al., 1999*; *Lonze et al., 2002*), specific inactivation of CREB in sympathetic neurons protects against developmentally regulated apoptosis (*Parlato et al., 2007*). Thus, the decrease in

**Video 4.** Impaired traffic of Rab5-GFP[+] endosomes in *Meis1* inactivated neurons. Representative videos showing 2 min acquisition of Rab5-GFP[+] endosomes trafficking in the axons of cultured *PLAT^CRE::Meis1^LoxP/LoxP* sympathetic neurons. See also *Figure 7*.

pCREB expression in *Meis1* mutant cannot explain the neuronal loss.

The increased apoptosis at E14.5 remains more intriguing. It involves about 5% of TH-positive cells in the SCGs and does not result in a significant volume decrease of the mutant SCGs at this stage. In other mutant mice, apoptosis at this stage has been reported (*Maina et al., 1998*; *Armstrong et al., 2011*). For instance, Met is expressed in SCG from E12.5 embryo (*Maina et al., 1998*) and blocking of HGF signaling or Met inactivation prevent early sympathetic neuroblasts differentiation into neurons from E12.5 to E14.5 without affecting their proliferation. During this period of development, HGF/Met favors the survival of sympathetic neuroblasts and in Met mutants apoptosis is increased at E14.5 (*Maina et al., 1998*). In Fz3 mutants, apoptosis is increased at E13.5 and E14.5 in sympathetic neuroblasts but cannot be rescued by *Bax* elimination (*Armstrong et al., 2011*). It is thus possible that Meis1 influences the survival of a small subset of sympathetic neuroblasts by interfering with one of the above pathways. Another explanation could be that a small subset of neurons depends on the retrograde transport of a trophic factor other than Ntrk1 at this stage. This hypothesis is consistent with the lack of induction of Syt1 expression at E14.5 whereas the induction of Ntrk1 expression is unaffected. Finally, our ChIP-seq experiment identified several potential target genes known to be involved in axonal growth and/or guidance (about 20%) as well as many other genes which function has not been yet studied. Thus, we cannot exclude that a deregulation of the expression of some of these genes is responsible for the apoptosis in this small subset of neurons. For instance, *Gdnf* was identified as a potential target gene in this search. Since the function of Gdnf on survival of early SCG neurons remains unclear, and because some studies detected Gdnf mRNA in embryonic sympathetic neurons (*Trupp et al., 1995*), a potential regulation of its expression by Meis1 could account for the increased apoptosis of this small subset of neurons.

To conclude, our results show that *Meis1* inactivation induces apoptosis of embryonic sympathetic neurons through different mechanisms: a severe loss of neurons occurring after E16.5 in which impaired retrograde trafficking of Ngf/Ntrk1 is a contributing factor together with other target-derived trophic factors, and a weaker apoptosis from E14.5 to E16.5 that is independent of Ngf/Ntrk1 retrograde signaling.

How *Meis1* is regulated in sympathetic neurons? In our study, we showed that *Meis1* is not regulated by Ngf. In several neuronal and non-neuronal tissues, Meis1 expression is differentially modulated by retinoic acid (RA) and bFGF. Mouse embryos treated with high doses of RA show increased levels of Meis1 (*Qin et al., 2002*) and in the chick limb bud, RA induces Meis1 expression whereas bFGF reduces it (*Mercader et al., 2000*). During development of striatonigral projection neurons, retinoic acid and FGF are required for positioning the boundaries of Meis1-expressing cells (*Rataj-Baniowska et al., 2015*). Thus, when RARβ is inactivated, the number of Meis1 positive cells is increased and this effect can be reversed by addition of FGF2 (*Rataj-Baniowska et al., 2015*). Meis1 expression is also induced in differentiated forebrain-derived neural stem cells in the presence of bFGF (*Barber et al., 2013*). In P19 and F9 carcinoma cells and in the U937 human leukemic monocyte lymphoma *cell* line, RA treatment rapidly induces Meis1 expression (*Ferretti et al., 2000*; *Longobardi and Blasi, 2003*; *Meester-Smoor et al., 2008*). In RA-treated human carcinoma cell line NT2/D1, a treatment that induces neuronal terminal differentiation, Meis1 is also up regulated (*Freemantle et al., 2002*). In agreement with a potential regulation of *Meis1* expression by RA during sympathetic neurons development, all RA receptors RARα, RARβ, RARγ and RXR are dynamically expressed (*Plum and Clagett-Dame, 1996*; *Doxakis and Davies, 2005*).

The role of RA in neurogenesis, axonal growth and maintenance of differentiated neurons is well known and relies on complex mechanisms depending on RA receptors expression, RA concentration, cellular context and time-exposure, (*Clagett-Dame and Plum, 1997*; *Maden, 2007*). The complexity of RA signaling is outlined by the various effects reported depending on embryonic stages and the species analyzed with sometimes conflicting results. In chick, a role of RA in sympathetic target field-innervation and neurons survival has been suggested (*Rodriguez-Tébar and Rohrer, 1991*; *von Holst et al., 1995*; *Plum and Clagett-Dame, 1996*; *Repa et al., 1996*; *Reis et al., 2002*). RA induces Ntrk1 expression and regulates the onset of Ngf dependence (*Rodriguez-Tébar and Rohrer, 1991*; *von Holst et al., 1995*; *Plum et al., 2001*). In chick, the expression of the other neurotrophins receptors is not modified by RA (*von Holst et al., 1995*), although RA also potentiates the survival and neurite outgrowth induced by Ntf3 (*Plum et al., 2001*) likely through Ntrk1 activation. At later stages, RA also reduces the survival effect of Gdnf and Neurturin of chick sympathetic neurons by preventing GFRα1, GFRα2 and Ret expression (*Doxakis and Davies, 2005*).

In rodents, the scenario seems different. RA treatment of mouse and rat sympathetic neurons prevents Ntrk1 mRNA developmental induction and increases Ntrk3, Ntrk2 and Ngfr mRNA expression (*Kobayashi et al., 1994*; *Wyatt et al., 1999*). Contrary to chick, RA enhances Gdnf and Ntf3 responsiveness of cultured rat sympathetic neurons (*Thang et al., 2000*). Interestingly, RA induces the expression of several synaptic vesicles proteins including SYT1 in human neural stem cell (*Ekici et al., 2008*). Less is known about the function of FGFs in early sympathetic neurons. FGF8 is expressed in E14.5 mouse sympathetic neurons (*Tanaka et al., 2001*), and FGF2 and FGFR1 are expressed in rat postmitotic sympathetic neurons (*Nindl et al., 2004*). bFGF induces a sympathetic neuron phenotype on PC12 cells and promotes neurite outgrowth and survival (*Rydel and Greene, 1987*). In chromaffin cells, it induces a dependence on Ngf for survival (*Stemple et al., 1988*). Thus, it seems that during sympathetic neurons development, RA and FGFs regulate the responsiveness of sympathetic neurons to multiple neurotrophic factors, which is in agreement with the function of Meis1 we propose. If such a regulation exists, it could be hypothesized that Meis1 is necessary to ensure the transition from local to target-derived neurotrophic signaling.

Two classes of transcription factors involved in late neuronal differentiation have been recently defined: terminal selector genes and dedicated maintenance factors (*Hobert, 2008*; *Hobert et al., 2010*; *Hobert, 2011*; *Deneris and Hobert, 2014*; *Allan and Thor, 2015*). Terminal selector genes group transcriptional regulators of effector genes defining a unique identity trait for a given subtype of neuron. Their removal results in the loss of many functionally unrelated terminal identity features such as neurotransmitter identity, neurotransmitter receptors, ion channels, whereas features shared by all neurons such as expression of synaptic vesicle proteins are unaffected (*Deneris and Hobert, 2014*). Dedicated maintenance transcription factors include genes whose expression commences after neuron-type identity is acquired and that are not involved in initial specification (*Deneris and Hobert, 2014*). Rather, they are necessary for maintenance of neuronal identity and survival. Terminal selector genes for sympathetic neurons unambiguously include *Phox2b*, *Gata3* and *Hand2* which the most studied function is to control the expression of TH and DBH, two key enzymes involved in the noradrenergic identity (*Rohrer, 2011*). Thus, according to these definitions and from our results, *Meis1* belongs to the second class of dedicated maintenance factors. A well-studied example of such factors is the Engrailed (Eng) family. In *Eng1/2* double mutants, mesencephalic dopaminergic neurons are generated and specified normally before being progressively eliminated by apoptosis (*Simon et al., 2001*, , *2004*; *Albéri et al., 2004*; ; *Sgado et al., 2006*). In vitro, proximal axonal growth in double *Eng1/2* mutant is unaffected whereas in vivo, the projections of these neurons toward the basal telencephalon are missing. Interestingly, in Eng1/Eng2 double mutants, neuronal apoptosis is preceded by a loss of α-synuclein expression (*Simon et al., 2001*), a vesicular protein able to modulate clathrin-dependent endocytosis, vesicle trafficking and recycling (*Kuwahara et al., 2008*; *Ben Gedalya et al., 2009*; *Nemani et al., 2010*; *Cheng et al., 2011*; *Kisos et al., 2014*; *Vargas et al., 2014*). Thus, the neuronal phenotypes we report following *Meis1* inactivation strikingly resemble the phenotypes reported for Eng mutation in mesencephalic dopaminergic neurons. Whether the regulation of generic traits for all neurons is a common feature of this class of dedicated maintenance factors remain to be explored.

Finally, it has been proposed that disruption of transcriptional maintenance programs is a key to increased susceptibility to late onset neurodegenerative disease (*Deneris and Hobert, 2014*). Since Meis1 is expressed by numerous post-mitotic neurons (*Toresson et al., 2000*), it is likely that the transcriptional program initiated by Meis1 that we reported in sympathetic neurons will be generalized to other class of CNS neurons.

## Materials and methods

### Animals

Procedures involving animals and their care were conducted according to European Parliament Directive 2010/63/EU and the 22 September 2010 Council on the protection of animals, and were approved by the institutional animal research committee (Departmental Directorate of protecting populations and animal health (ethics for animal welfare and environmental protection, N° A 34-485/CEEA-LR-12074) and by our Ethics committee for animal experiments, Languedoc Roussillon, N° 34–376, February the 17$^{th}$ of 2009). *Meis1$^{LoxP/LoxP}$*, *PLAT$^{CRE}$*, and *Bax$^{-/-}$* mice were generated as

previously described (*Knudson et al., 1995*; *Danielian et al., 1998*; *Pietri et al., 2003*; *Unnisa et al., 2012*). Primers used to genotype the different strains were: *Meis1* sense 5'-CCA AAG TAG CCA CCA ATA TCA TGA-3'; *Meis1* antisense 5'-AGC GTC ACT TGG AAA AGC AAT GAT-3'; *Bax* sense 5'-GAG CTG ATC AGA ACC ATC ATG-3'; *Bax* antisense 5'-GTT GAC CAG AGT GGC GTA GG-3'; *Neo* antisense 5'-CCG CTT CCA TTG CTC AGC GG-3'; *CRE* sense 5'-TGC CAG GAT CAG GGT TAA AG-3'; *CRE* antisense 5'-GCT TGC ATG ATC TCC GGT AT-3'. Mice were kept in an animal facility and gestational stages were determined according to the date of the vaginal plug. For ECG recordings, sex-matched 12-weeks old transgenic and littermates mice (n = 8 in each group) were used.

## Antibodies

The antibodies used are: mouse anti-Meis1 (Millipore, 05–779, 1/100 for immunoblotting, Germany), goat anti-Meis1 (Santa Cruz, sc-10599, 1/100 for immunochemistry, Dallas, Texas, USA), sheep anti-TH (Thermo Scientific, PA1-4679, 1/2000, Waltham, MA, USA), rabbit anti-PGP9.5 (Millipore, AB1761, 1/500, Germany), goat anti-Nrp-1 (R&D Systems, AF566, 1/250, Minneapolis, USA), rabbit anti-Synaptophysin (Cell Signaling Technology, 5461, 1/250, Danvers, MA, USA), rabbit anti-Phox2b (kindly provided by JF Brunet, 1/250, or Abcam ab183741, 1/250, UK), goat anti-c-Ret (R&D Systems, AF482, 1/20000, Minneapolis, USA), rabbit anti-Ntrk1 (Millipore, 06–574, 1/250, Germany), rabbit anti-Cleaved Caspase-3 (Cell Signaling Technology, 9661, 1/250, Danvers, MA, USA), rabbit anti-Clathrin Heavy Chain 1 (Thermo Scientific, PA5-17347, 1/200, Waltham, MA, USA), Phospho-CREB (Cell Signaling, 87G3, 1/200, Germany), Rab5 (Abcam, ab18211, 1/200, UK), Sox2 (Santa Cruz, sc-17320, 1/250, Dallas, Texas, USA).

## Immunochemistry

Immunofluorescent staining on frozen sections were performed as previously described (*Marmigère et al., 2006*). For Phox2b and Meis1 immunostaining, an epitope retrieval step was carried out by immersion of the sections for 15 min at 68° in sodium citrate buffer (10 mM Sodium Citrate, 0.05% Tween 20 [pH 6]).

To analyze sympathetic neurons proliferation, intraperitoneal injections of 0.7 mg of EdU (5-ethynyl-2′-deoxyuridine; A10044, Life Technologies) were performed on E16 pregnant mice 3 hr before sacrifice. Embryos were harvested and tissues sections were prepared as previously described. Incorporated EdU was revealed using the Click-iTEdUAlexa Fluor488 Imaging Kit (Life Technologies, C10337, Waltham, MA, USA), according to the manufacturer's instructions. Wide field microscopy (Leica DMRB, Germany) was only used for ISH images and confocal microscopy (Leica SP5-SMD, Germany) for immunochemistry except for the *Figure 4—figure supplement 1* where an epifluorescent microscope (Leica DMRB, Germany) was used. Confocal images are presented as single stacks except for images in *Figure 3B* that is displayed as maximal projections.

## Hematoxylin-Eosin staining

Air dried frozen sections were washed in water then stained with hematoxylin for 1 min at room temperature and washed extensively with water. Slides were then dehydrated in a PBS/alcohol (70%) solution and stained with eosin for 30 sec at room temperature. After serial wash in water, sections were finally dehydrated in PBS solutions with increasing alcohol concentration (50%, 75%, 95%, and 100%), mounted and observed with a microscope (Leica DMRB, Germany).

## Transmission electron microscopy (TEM)

For electron microscopy, SCGs were dissected with their exiting nerve and immersed in a solution of 2.5% glutaraldehyde in PHEM buffer (1X, pH 7.4) overnight at 4°C. They were then washed in PHEM buffer and post-fixed in a 0.5% osmic acid + 0.8% *Potassium ferrocyanide solution* for 2 hr at room temperature in the dark. After two washes with PHEM buffer, samples were dehydrated with solutions of increasing ethanol concentration (30–100%). Samples were embedded in EmBed 812 using an Automated Microwave Tissue Processor for Electronic Microscopy, Leica EM AMW. Thin sections (70 nm; Leica-Reichert Ultracut E) were collected at different levels of each block. These sections were counterstained with uranyl acetate and lead citrate and observed using a Hitachi 7100 transmission electron microscope in the Centre de Ressources en Imagerie Cellulaire de Montpellier (France).

## Probes for in situ hybridization (ISH)

RNA probes were generated from cDNA sequences that were PCR-amplified from reverse-transcribed total RNA isolated from whole E13 mouse embryos. Amplified fragments were cloned into the pGEM-T easy vector using the TA cloning kit (Promega, Madison, Wisconsin, USA) and confirmed by sequencing. Ntrk1, Ntrk2, Ntrk3, Ret, DBH, Gata3, Hand2, Sox10 probes were published elsewhere (*Kuhlbrodt et al., 1998*; *Tsarovina et al., 2004*; *Bourane et al., 2007*). Meis1 probe was a generous gift from Mark Featherstone (*Shanmugam et al., 1999*). Digoxigenin- (DIG) and fluorescein- labeled RNA probes were synthesized using the DIG- or Fluorescein labeling kit (Roche, Switzerland), respectively, according to the manufacturer's instructions. Simple and double ISH were conducted as previously described (*Marmigère et al., 2006*). The primers used for cloning ISH probes are indicated in the following table.

|        | Forward | Reverse |
|--------|---------|---------|
| Meis2  | 5'-ATGGCGCAAAGGTACGATGAGCT-3' | 5'- TTACATATAGTGCCACTGCCCATC-3' |
| Meis3  | 5'-ATGGCCCGGAGGTATGATG-3' | 5'-CTATAGGTAATGCCACTCTCCT-3' |
| Ngfr   | 5'-AGC GGCATCTCTGTGGAC-3' | 5'-GGAGAAGGGAGGGGTTGA-3' |
| Adam19 | 5'-GTTTTACCGCTC CCTGAACA-3' | 5'-GGAAGATTC AGT GCCAGAGC-3' |
| Nbea   | 5'-GAGCAGAGTTTTGCCCACTC-3' | 5'-CATAGCATGGCACAACAACC-3' |
| Dst    | 5'-AGTGAACAGAAACCCGTTGG-3' | 5'-TCAGCATGTTGTCCAGCTTC-3' |
| Syt1   | 5'-CGATGCTGAAACTGGACTGA-3' | 5'-AAGGGCATAGGGGCTTTCTA-3' |
| Lnp    | 5'-CCCAGCGCTCTATCTGTAGG-3' | 5'-GTTGAGCTTGAGCCTTCCAC-3' |
| Klf7   | 5'-ACAAAACAAAACAAAAGGGCCACTG-3' | 5'-CAGGTGCAAAGCCCTTTAAG-3' |

## In situ hybridization (ISH)

RNA probes were generated from cDNA sequences that were PCR-amplified from reverse-transcribed total RNA isolated from whole E13 mouse embryos using primers listed in expanded view. Amplified fragments were cloned into the pGEM®-T easy vector using the TA cloning kit (Promega) and confirmed by sequencing. Ntrk1, Ntrk2, Ntrk3, Ret, DBH, Gata3, Hand2, Sox10 probes were published elsewhere (*Kuhlbrodt et al., 1998*; *Tsarovina et al., 2004*; *Bourane et al., 2007*). Meis1 probe was a generous gift from Mark Featherstone (*Shanmugam et al., 1999*). Digoxigenin- (DIG) and fluorescein- labeled RNA probes were synthesized using the DIG- or Fluorescein labeling kit (Roche, Switzerland), respectively, according to the manufacturer's instructions. Simple and double ISH were conducted as previously described (*Marmigère et al., 2006*).

## Cell culture and live imaging

SCG explants were dissected from E16 mouse embryos and seeded in FluoroDish dishes (World Precision Instruments, Sarasota, FL, USA) coated with poly-L-ornithine (0.5 mg/ml) and laminin (5 µg/ml). Explants were cultivated in neurobasal medium (Life Technologies, Waltham, MA, USA) containing B27 serum (2%, Life Technologies, Waltham, MA, USA), Glutamine (2 mM, Life Technologies, Waltham, MA, USA), Glutamic acid (25 µM, Life Technologies, Waltham, MA, USA), Penicillin-Streptomycin (200 U/ml, Life Technologies, Waltham, MA, USA) and Ngf (10 ng/ml, Life technologies, MNAC-25, Waltham, MA, USA). After 24 hr in culture, explants were electroporated with a Rab5-GFP plasmid (*Roberts et al., 1999*) (500 ng/µl) in PBS 1X using a BTX *Electro Square Porator ECM 830* (2 times 3 pulses of 200 V/cm, separated by intervals of 15 ms). The movements of Rab5-GFP[+] vesicles were recorded 24h after electroporation on a confocal microscope (Leica SP5-SMD, Germany) at 37°C in a *humidified* atmosphere containing 5% $CO_2$ for 2 min for each acquisition.

## Western blot

Cells were lysed in Laemmli buffer containing 62.5 mM Tris-HCl, pH 6.8, 25% glycerol, 2% SDS, 0.01% Bromophenol Blue and 5% β-mercaptoethanol. All protein samples were separated on SDS-

PAGE, transferred on nitrocellulose membrane (Proteigene, 10401197, France), and probed with primary antibodies. HRP-conjugated goat anti-mouse secondary antibody (Jackson ImmunoResearch, 115-035-174, 1/500, Baltimore, PA, USA) was revealed by chemo-luminescence using the Pierce ECL Plus Western Blotting Substrate (Thermo Scientific, 32132, Waltham, MA, USA). Signals were detected using autoradiographic films (Sigma Aldrich, Z363022-50EA, Saint-Louis, Missouri, USA) or light sensitive camera (BioRad, ChemiDo XRS + System, Hercules, California, USA).

## Quantifications

For the calculation of innervation density, the total TH/PGP9.5 fibers areas were measured with ImageJ software (*Schneider et al., 2012*), using the "'analyze particles' option after adjusting the threshold with the max entropy method. The total neurites length was measured with ImageJ software on multiple random fields for each experimental condition. The calculation of SCGs volumes, immunochemistry intensities and ISH intensities were measured with the ImageJ software using the 'measure'option. The intensity of the background signal was subtracted to each measure. All quantifications were realized on at least 3 sections per embryos and at least 3 different embryos per genotype. For SCG volumes calculation, the area of the SCG present in each section within a series were measured, increased by the number of series and the thickness of the sections, and finally added. Statistical comparisons were performed by a Student's T-test. Values and errors bars indicate respectively the mean and the s.e.m.

In live imaging experiment, time-lapse acquisition was performed on 6 to 8 different neurons from 3 different WT and $PLAT^{CRE}::Meis1^{LoxP/LoxP}$ SCG explants.

In TEM experiments, the number of endocytic structures per axons and the area ratio (area of endocytic structures per axons/area of axons) were measured on at least 5 pictures randomly taken in each samples. Samples were obtained from 3 WT and 2 $PLAT^{CRE}::Meis1^{LoxP/LoxP}$ sympathetic nerves.

## ECG recording and HRV (Heart Rate Variability) analysis

Electrocardiogram (ECG) monitoring (2000 Hz) were performed by telemetry system using Physio-Tel®ETAF20 teletransmitter (DSI, USA) and IOX software (EMKA Technologies, France). Continuous 24 hr ECGs were analyzed with ECG-auto software (EMKA Technologies, France), in respect of circadian light/dark cycles (12 hr/12 hr). The mean RR interval and the mean PR, QRS, QTc durations were calculated. The QT interval was defined as the time between the first deviation from an isoelectric PR interval until the return of the ventricular repolarization to the isoelectric TP baseline from lead II ECGs. The QT correction was performed with the adapted Bazett's formula of Mitchell (*Mitchell et al., 1998*). Arrhythmia were detected and counted by hand, according to the Lambeth conventions guidelines (*Walker et al., 1988*). ECG analyses were also performed after pharmacological i.p. injection of isoproterenol hydrochloride (β-adrenoreceptor agonist, 1 mg.kg$^{-1}$, NaCl 0.9%) or carbamoylcholine chloride (non-selective cholinergic agonist, 0.5 mg.kg$^{-1}$, NaCl 0.9%). The chronotropic response was studied during exercise on treadmill (10 min of baseline recording, following by 5 min at 90 m.min$^{-1}$, 2 min at 120 m.min$^{-1}$, 2 min at 140 m.min$^{-1}$, 2 min at 160 m.min$^{-1}$ and 6 min of recuperation).

Heart rhythm is triggered by electrical activity which is continuously adapted to metabolic needs. This beat-to-beat regulation is largely modulated by the autonomic nervous system (ANS). ANS controls the heart rhythm by modifying the automatic sinus activity through a complex interplay of the orthosympathetic and parasympathetic (or vagal) systems. The orthosympathetic system increases beat delivery via the rostral ventrolateral medulla whereas the vagal system slows it via the ambiguus and vagal dorsal motor nuclei. Time- and frequency domain indices of HRV are the standard parameters to evaluate ANS activity as well in clinics as in fundamental research. Time-domain analysis measures changes in R–R intervals between successive normal (sinusal) cardiac cycles over time. All HRV indices in the time domain are based on descriptive statistical calculations. In frequency domain, power spectral analysis is performed by Fast Fourier Transform, a non-parametric method characterized by discrete peaks for the several frequency components (*Akselrod et al., 1981*; *Malliani et al., 1994*; *Task Force of the European Society of Cardiology the North American Society of Pacing Electrophysiology 1996*). The FFT thus converts time information into frequency information by decomposing the periodic oscillations of sinus heart rate into harmonics

characterized by specific frequencies and amplitudes. The power spectrum of HRV in mice is similar those derived from humans showing two principal components: the LF and HF frequencies (*Gehrmann et al., 2000*; *Thireau et al., 2008*). In this study, HRV was evaluated by power spectra analysis ($ms^2$) using the fast Fourier transformation (segment length of 2048 beats, linear interpolation with resampling to a 20-Hz interbeat time series and Hamming windowing). The cut-off frequency ranges for the low frequency (LF: 0.15–1.5 Hz) and high frequency powers (HF: 1.5–5 Hz) were chosen according to those used in the literature (*Thireau et al., 2008*). As in humans, the low frequency reflects a complex interaction between sympathetic and parasympathetic ways that modulates heart rate in mice (*Gehrmann et al., 2000*; *Thireau et al., 2008*). Therefore, we assessed total variability with the standard deviation of all normal RR intervals (SDNN) and, the vagal activity from HF (*Gehrmann et al., 2000*) in the frequency domain and from the square root of the mean square successive differences between successive normal intervals (RMSSD) in the time domain. The cardiac sympathetic and baroreflex activities were assessed from LF (*Electrophysiology, 1996*; *Malliani, 1999*; *Gehrmann et al., 2000*; *Thireau et al., 2008*) in the frequency domain. As in previous studies, we used the LF to HF ratio (LF/HF) to quantify the sympathovagal balance (*Berul et al., 2000*; *Tankersley et al., 2004*). In mice the sympathetic nervous system is one major factor that controls basal heart rate (*Mansier et al., 1996*; *Uechi et al., 1998*).

## Chromatin immuno-precipitation (ChIP)

Chromatin immune-precipitation has been performed with the Low Cell ChiP Kit (Diagenode, kch-maglow-A16, Belgium) according to the manufacturer's instructions with minors modifications. Eighteen to 25 SCGs from E16 C57Bl6 embryos were dissected in PBS with protease inhibitors (Sigma-Aldrich, P2714-1BTL, Saint-Louis, Missouri, USA). Fixation was done by adding PBS/PFA 1% for 10 min under agitation at room temperature and then stopped in 0.125 µM glycine for 5 min under agitation at room temperature. After three washings in PBS1X, chromatin was isolated and sheared by sonication (16 cycles composed by 30' pulses following by 30' without pulses on a BIORUPTO-R STANDARD) to obtain a smear between 250 and 1000 bp. Immunoprecipitation was performed overnight at 4°C with 10 µL of beads previously coupled with 10 µL of Meis1/2/3 antibody (Millipore, 05–779, Germany) or IgG1 isotype control antibody (Sigma-Aldrich, M5284). The Meis1/2/3 antibody has been previously validated for ChIP experiment or co-immunoprecipitation (*Shim et al., 2007*; *Pfeufer et al., 2010*). The DNA was then purified using the iPure kit (Diagenode, C03010012, Belgium) according to the manufacturer's instructions. Sequencing was realized by MGX platform (Montpellier Genomix, Montpellier) using a HiSeq 2000 sequencing system (Illumina, San Diego, California, USA). Data were analyzed using the CA-SAVA software for alignment and the MACS software for statistical analyses. Signals were annotated by the name of the genes flanking the 20 kb to the precipitated sequence. Bibliographical search was realized using PubMed. The data discussed in this publication have been deposited in NCBI's Gene Expression Omnibus (*Edgar et al., 2002*) and are accessible through GEO Series accession number GSE54144 (http://www.ncbi.nlm.nih.gov/geo/query/acc.cgi?acc=GSE54144).

## Echocardiography

Doppler echocardiography was performed using a high resolution ultrasound system (Vevo 2100; VisualSonics, Toronto, Canada) equipped with a 40-MHz transducer. Mice were anesthetized with 1.5% isofluranein 100% oxygen to reach comparable heart rate and placed on a heating table in a supine position. Body temperature was monitored through a rectal thermometer to be maintained at 36–38°C and ECG was recorded all along the echocardiographic procedure with limb electrodes. Ejection fraction (EF%) and fractional shortening (FS%) were calculated from the left ventricular diameters on M-mode measurements at the level of papillary muscles in a parasternal short-axis two dimensional view. To better consider left ventricular morphology, EF was also calculated from a B-mode parasternal long axis view (EF% B-mode) by tracing endocardial end-diastolic and end-systolic borders to estimate left ventricular volumes, and the endocardial fractional area change (FAC%) on a parasternal short-axis view at papillary muscle level was similarly measured. Mitral flow was recorded by a pulsed-wave Doppler sampling at the tips of the mitral valves level from the apical four-chamber view. Peak early (E) and late atrial contraction (A) mitral inflow waves velocities were measured and the ratio E/A was calculated. Pulsed-wave Doppler of the ascending aortic blood flow was

recorded permitting measurements of the velocity time integral (AoVTI). All measurements were quantified and averaged for three cardiac cycles.

## Acknowledgements

This work was supported by Inserm Avenir grant, AFE grant (Association France-Ekbom) and Fondation de France (Synaptocard project, N°2013-00038586). FM, SR, JT hold CNRS positions and FB was supported by the University of Montpellier2. We thank S Dufour for generous gift of the PLAT-$^{CRE}$ strain, JF Brunet for the Phox2b antibody, MS Featherstone for the mouse *Meis1* cDNA, A Pattyn for the Hand2, Gata3, DBH probes vectors, CD Deppmann for *Ngf* KO embryos, C Cazevieille for help with TEM, S Salinas for Rab5-GFP plasmid and L Journot and the MGX (Montpellier Genomix) platform for advices on the ChIP-seq experiments. We also thank P de Santa Barbara, C Raoul, P Carroll and A Pattyn for helpful discussions and critical reading of the manuscript. NGC and NAJ are supported by the Cancer Prevention and Research Institute of Texas (CPRIT). They are also both CPRIT Scholar's in Cancer Research.

## Additional information

### Funding

| Funder | Grant reference number | Author |
| --- | --- | --- |
| Association France Ekbom | Grant R13018FF | Jérôme Thireau<br>Yves Dauvilliers<br>Sylvain Richard<br>Frédéric Marmigère |
| Fondation de France | Synaptocard 2013-00038586 | Jérôme Thireau<br>Charlotte Farah<br>Sylvain Richard |
| Fondation de France | Synaptocard N$_{\deg}$ 2013-00038586 | Jérôme Thireau<br>Yves Dauvilliers<br>Sylvain Richard<br>Frédéric Marmigère |
| Institut National de la Santé et de la Recherche Médicale | Avenir program | Frédéric Marmigère |

The funders had no role in study design, data collection and interpretation, or the decision to submit the work for publication.

### Author contributions

FB, CF, Conception and design, Acquisition of data, Analysis and interpretation of data, Drafting or revising the article; JT, FM, Conception and design, Acquisition of data, Analysis and interpretation of data, Drafting or revising the article, Contributed unpublished essential data or reagents; SV, SK, Conception and design, Acquisition of data, Drafting or revising the article; YD, JV, Conception and design, Analysis and interpretation of data, Drafting or revising the article; NGC, NAJ, SR, Conception and design, Analysis and interpretation of data, Drafting or revising the article, Contributed unpublished essential data or reagents

### Author ORCIDs

Frédéric Marmigère, http://orcid.org/0000-0002-0515-7483

### Ethics

Animal experimentation: Procedures involving animals and their care were conducted according to European Parliament Directive 2010/63/EU and the 22 September 2010 Council on the protection of animals, and were approved by the institutional animal research committee (Departmental Directorate of protecting populations and animal health (ethics for animal welfare and environmental protection, N° A 34- 485/CEEA-LR-12074) and by our Ethics committee for animal experiments, Languedoc Roussillon, N° 34-376, February the 17th of 2009).

## Additional files

### Major datasets

The following datasets were generated:

| Author(s) | Year | Dataset title | Dataset URL | Database, license, and accessibility information |
|---|---|---|---|---|
| Fabrice Bouilloux, Frédéric Marmigère | 2015 | Search for Meis1 target genes in embryonic sympathetic neurons | http://www.ncbi.nlm.nih.gov/geo/query/acc.cgi?acc=GSE54144 | Publicly available at the Gene Expression Omnibus (accession no. GSE54144) |

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
