## [Decision Letter]

Thank you for submitting your work entitled "Loss of *Meis1* prevents sympathetic neurons target-field innervation and increases susceptibility to sudden cardiac death" for consideration by *eLife*. Your article has been reviewed by three peer reviewers, and the evaluation has been overseen by David Ginty (Reviewing Editor) and K VijayRaghavan (Senior Editor).

The reviewers have discussed the reviews with one another and the Reviewing editor has drafted this decision to help you prepare a revised submission.

Summary:

In this manuscript, Bouilloux et al., propose a role for the transcription factor, *Meis1*, in neurotrophin-dependent development of sympathetic neurons, likely via regulating the vesicular machinery. There are several strengths of this study; (1) the phenotypes in the *Meis1* mutant mice, specifically the defects in autonomic physiology and the developmental defects in the sympathetic nervous system are striking and convincingly demonstrated. Specifically, the autonomic physiology defects in sympathetic regulation of cardiac functions are interesting and highlight the consequences of perturbations in peripheral innervation during development on target organ function later in life. (2) As noted by the authors, much of the work on transcriptional regulation of sympathetic nervous system development has been restricted to early stages of neuronal specification and differentiation. Thus, the identification of a transcription factor that influences later aspects of sympathetic neuronal survival and connectivity is a significant aspect of this study. The study does a nice job demonstrating physiological relevance with the heart analysis and the modeling of sudden cardiac death syndrome. Overall this is a rigorous and fairly comprehensive analysis. (3) In general, the physiological and histochemical analyses have been done rigorously and the data are of high quality. The reviewers appreciate the data showing that cell death in *Meis1* sympathetic ganglia is prevented by removing *Bax*, similar to that noted in *NGF;Bax* double knockout mice. This supports the authors’ conclusions that it is post-mitotic sympathetic neurons that rely on target-derived neurotrophins that are being eliminated in the absence of *Meis1*, and not neuronal precursors or glial cells.

Essential revisions:

1) There is a serious concern about the mechanistic data and the interpretations. Specifically, whether the developmental phenotypes are due to a failure in retrograde NGF signaling is not clear. Concerns include: i) innervation of some targets (e.g. trachea) is strongly reduced in *HtPACRE/Meis1^LoxP/LoxP^*mice but not in *NGF^-/-^* mice (Glebova and Ginty, 2004) and, ii) the early cell death at E14.5 (Figure 3) cannot be explained by reduced NGF/TrkA signaling. Naturally occurring cell death in sympathetic ganglia starts at E16-17 (Coughlin and Collins, 1985). In mice that lack NGF neuron loss can be detected by E17.5 (Crowley at al., 1994; Francis and Collins, 1999), iii) although pCREB may indicate NGF signaling it should be noted that sympathetic neuron development is not dependent on cell-autonomous CREB/pCREB (Parlato et al, 2007). Thus, while a deficit in retrograde NGF signaling may be a contributing factor it is unclear whether it accounts for the phenotype. There is also the technical concern that the immunostaining data for clathrin, Rab5 and synaptotagmin (Figure 6) are not convincing enough to make the claim that "clathrin-coated vesicles are virtually absent from *Meis1* inactivated neurons" (Discussion section). Thus, the reviewers agree that your study requires direct analysis of TrkA trafficking (TrkA internalization and axonal transport) and more rigorous imaging of endocytic/synaptic organelles (instead of just Rab5 transport). Also, even if deficits in NGF trafficking are observed, statements in the Abstract and Discussion that disrupted retrograde NGF signaling accounts for the sympathetic neuron survival defects should be softened.

2) The human tissue plasminogen activator (HtPA) Cre-mouse targets all neural crest derivatives including heart outflow tract (Pietri et al, 2003). As the *Meis1*-knockout has defects in the heart outflow tract (Stankunas et al., 2008) it may be expected that the development of cardiac neural crest derivatives is also affected in the *HtPACRE/Meis1^LoxP/LoxP^*mice. The authors argue that heart morphology and function are normal in *HtPACRE/Meis1^LoxP/LoxP^*mice (e.g. from Figure 1—figure supplement 2). In the Discussion (first paragraph) the authors refer to Mahmoud et al. (2013), which showed that a conditional *Meis1* knockout in cardiomyocytes does not affect heart morphology and physiology. But HtPACRE is acting earlier and on different cell types than aMHCCRE. This is a potential complication and some of the conclusions would be in question if effects on heart development rather than sympathetic neuron development cannot be excluded in *HtPACRE/Meis1^LoxP/LoxP^* mice. The reviewers think that a more detailed explanation of the heart defects in the *Meis1* mutant mice is warranted, in the Discussion.

3) The authors should provide data on the innervation density in adult hearts, preferably of mice that show the described physiological phenotype to exclude the possibility that the surviving neurons have sprouted between E16.5 and adult stages.

---

## [Author Response]

*Essential revisions:*

*1) There is a serious concern about the mechanistic data and the interpretations. Specifically, whether the developmental phenotypes are due to a failure in retrograde NGF signaling is not clear. Concerns include: i) innervation of some targets (e.g. trachea) is strongly reduced in HtPACRE/Meis1^LoxP/LoxP^ mice but not in NGF^-/-^ mice (Glebova and Ginty, 2004) and, ii) the early cell death at E14.5 (Figure 3) cannot be explained by reduced NGF/TrkA signaling. Naturally occurring cell death in sympathetic ganglia starts at E16-17 (Coughlin and Collins, 1985). In mice that lack NGF neuron loss can be detected by E17.5 (Crowley at al., 1994; Francis and Collins, 1999), iii) although pCREB may indicate NGF signaling it should be noted that sympathetic neuron development is not dependent on cell-autonomous CREB/pCREB (Parlato et al, 2007). Thus, while a deficit in retrograde NGF signaling may be a contributing factor it is unclear whether it accounts for the phenotype. There is also the technical concern that the immunostaining data for clathrin, Rab5 and synaptotagmin (Figure 6) are not convincing enough to make the claim that "clathrin-coated vesicles are virtually absent from Meis1 inactivated neurons" (Discussion section). Thus, the reviewers agree that your study requires direct analysis of TrkA trafficking (TrkA internalization and axonal transport) and more rigorous imaging of endocytic/synaptic organelles (instead of just Rab5 transport). Also, even if deficits in NGF trafficking are observed, statements in the Abstract and Discussion that disrupted retrograde NGF signaling accounts for the sympathetic neuron survival defects should be softened.*

We fully understand the concern of the reviewers. It was not our intention in the first place to leave the reader with the impression that impaired retrograde NGF/TrkA signaling was responsible alone for all the defects we report in *Meis1* mutant. Several changes have been made throughout the manuscript in the summary, the presentation of the results and Discussion to emphasize this essential point and we hope that the revised version has now cleared up this ambiguity (Abstract; subsection “Loss of target-field innervation signaling pathways in *Meis1*-inactivated sympathetic neurons”, last paragraph; Discussion, seventh and eighth paragraphs). We now emphasize that the early apoptosis in *Meis1* mutant (E14.5) is not compatible with the period of target-derived NGF dependent apoptosis, and that whereas the late apoptosis (E16.5-18.5) likely involves NGF/TrkA retrograde signaling, the default in the innervation of the trachea indicates that other signaling pathways must be affected. In the Results section, we now describe the progressive loss of neurons as occurring in two phases (subsection “*Meis1* deficient sympathetic neurons progressively die by apoptosis”, last paragraph, Discussion, eighth paragraph), with one clearly independent of NGF/TrkA retrograde signaling, and a second one during which the neuronal loss is accentuated and to which NGF/TrkA retrograde signaling contributes. In the Discussion, we now discuss which additional pathways could be involved (paragraphs eight to ten). We also cite the manuscript of Parlato et al. (Parlatoet al., 2007) to indicate that the loss of pCREB expression cannot account for apoptosis, and we emphasize that pCREB staining was only used as readout of retrograde NGF/TrkA signaling (subsection “Loss of target-field innervation signaling pathways in *Meis1*-inactivated sympathetic neurons“; Discussion, eighth paragraph).

We also agree that the decreased immunostaining for clathrin, Rab5 and synaptotagmin (Figure 6) are not convincing enough to make the claim that "clathrin-coated vesicles are virtually absent from *Meis1* inactivated neurons". This sentence has been changed to “clathrin immunoreactivity is virtually absent from *Meis1* inactivated neurons" (Discussion). In addition, we bring a new set of Transmission Electron Microscopy (TEM) analysis of E16.5 sympathetic nerves. In these analyses, we observed a decrease in the surface area occupied by endocytic figures, as well as the number of these endocytic figures per surface area of axons. These TEM results are now presented in Figure 6—figure supplement 2. Six to seven embryonic sympathetic nerves from different embryos in each group were dissected and prepared for TEM. However, because of the time needed for dissection combined with a critical fixation time, we were satisfied with the quality of only 2 to 3 of these preparations in each group. For this reason, and because according to experts, TEM is difficult to quantify, these results are presented in a supplementary figure and were not statistically analyzed. Altogether, our results on clathrin, syp1 and Rab5 immunoreactivity along with Rab5-GFP overexpression and TEM analysis indicate that not only immunoreactivity is decreased but also the number and size of endocytic figures.

During the time of revision, we carried out several attempts to directly analyze TrkA trafficking in our explant culture of SCG neurons. Two different TrkA-GFP constructs from different sources were used. Unfortunately and for unknown reasons, we were not able to detect TrkA-GFP following electroporation even in WT cultures. A reason might be the much larger size of the TrkA-GFP plasmid construct compared to the Rab5-GFP construct. Thus, in our conditions, these TrkA-GFP plasmid constructs were not efficiently transfected. These different trials brought our colony so low that several months now would be needed to properly set-up the experimental conditions. We feel very sorry that we cannot bring a positive answer on this special point. Nevertheless, as mentioned by the reviewer, even with a direct visualization of TrkA trafficking, statements in the abstract and discussion should have been softened which was done throughout the entire manuscript.

*2) The human tissue plasminogen activator (HtPA) Cre-mouse targets all neural crest derivatives including heart outflow tract (Pietri et al, 2003). As the Meis1-knockout has defects in the heart outflow tract (Stankunas et al., 2008) it may be expected that the development of cardiac neural crest derivatives is also affected in the HtPACRE/Meis1^LoxP/LoxP^ mice. The authors argue that heart morphology and function are normal in HtPACRE/Meis1^LoxP/LoxP^ mice (e.g. from Figure 1—figure supplement 2). In the Discussion (first paragraph) the authors refer to Mahmoud et al. (2013), which showed that a conditional Meis1 knockout in cardiomyocytes does not affect heart morphology and physiology. But HtPACRE is acting earlier and on different cell types than aMHCCRE. This is a potential complication and some of the conclusions would be in question if effects on heart development rather than sympathetic neuron development cannot be excluded in HtPACRE/Meis1^LoxP/LoxP^ mice. The reviewers think that a more detailed explanation of the heart defects in the Meis1 mutant mice is warranted, in the Discussion.*

We thank the reviewers for pointing out this point. In view of the published works on cardiac septum defect in *Meis1* full knockout (Stankunaset al., 2008) and the well-known participation of neural crest cells in the formation of the cardiac septum, we understand the concern of the reviewers whether or not structural cardiac defects are present in the *HtPA^Cre^/Meis1^LoxP/LoxP^*strain and contribute to the reported phenotype. To solve this issue, we now added some new data and extended the discussion of our echocardiographic and ECG analysis.

Histological analysis of *HtPA^Cre^/Meis1^LoxP/LoxP^*E14.5 embryos and control littermates revealed no septal defect in any of the analyzed embryos. To better understand the difference in septal defect between full *Meis1* KO and *HtPA^Cre^/Meis1^LoxP/LoxP^* mice, we generated *Wnt1^CRE^/ Meis1^LoxP/LoxP^* animals. The *Wnt1^CRE^* also induces recombination in neural crest cells and their derivatives (to our experience, earlier and broader) and has previously been used to inactivate genes in cardiac neural crest derivatives. Because the *Wnt1^CRE^* transgene is inserted near the *Meis1* locus on chromosome 11, the extremely low rate of recombination prevented a large study and only one *Wnt1^CRE^/Meis1^LoxP/LoxP^* mutant could be generated out of 120 embryos collected (although Mendelian ratios were consistent). In this embryo, there was a septal defect similar to the one described in other *Meis1* KO strains. This difference means that either the recombination efficiency is low in neural crest cells contributing to the cardiac lineage when using the *HtPA^CRE^* line or that *Meis1* is important earlier during cardiac neural crest cells delamination because the Wnt1^CRE^ strain induces recombination earlier than the *HtPA^CR^*^E^ strain. These results are now presented (Figure 1—figure supplement 1), described in the Results section (subsection “*Meis1* inactivation in the PNS compromises the sympatho-vagal regulation of cardiac function”, second paragraph), and discussed in the Discussion section (second paragraph). Accordingly, whereas the *Wnt1^CRE^*strain has been used widely to study cardiac neural crest cells, we were unable to find any report using the *HtPA^CRE^* strain to study this lineage although it has been used in several reports studying the peripheral nervous system including enteric, sympathetic and sensory neurons (Pietriet al., 2003, Pietriet al., 2004, Breauet al., 2006, Haoet al., 2006). These results strongly vindicate our choice of the *HtPA^CRE^* strain to study *Meis1* function in the peripheral sympathetic nervous system and the consequences of its inactivation on autonomic regulation of cardiac functions. We conclude that the cardiac phenotype we report in the present study is not complicated by the morphological heart defects previously reported by Stankunas et al.

This conclusion is strengthened by the morphologic, functional and hemodynamics parameters measured in our transthoracic-echocardiography experiments. According to the guidelines recommendations for left ventricle function assessment (Langet al., 2015), measurements performed in the parasternal long axis and short axis views allow characterizing contractile function and morphology. Both views attest to the absence of wall structure defects in the *Meis1* mutant, including the septal wall observed in the long axis view. The ejection fraction (EF), which is used as the conventional contractile function index, was not different from WT mice. Measuring systolic performance tracing all along endocardial end-diastolic and end-systolic borders also excludes a possible left ventricular abnormal regional remodeling (EF B-mode and FAC; [Supplementary-material SD1-data]). Finally, heart diastolic performance was assessed by measuring left ventricle filling waves (Figure 1—figure supplement 2/A ratio in [Supplementary-material SD1-data]) in standard 4 cavities view. A septum closure defect would result in different filling values in WT and mutant mice because of hemodynamic and rigidity consequences. To conclude, in our *HtPA^CRE^/Meis1^LoxP/LoxP^*mice, basal septal wall observation during 4 cavities view and evaluation of aortic and ventricular hemodynamic outflows measured functionally by echocardiography did not revealed any impairment.

In addition, ECG analyses during baseline did *not show* alterations reflecting structural abnormalities, and ECG analyses during pharmacological challenge with compounds that mimic autonomic neurotransmitters showed that only the regulation of cardiac function by the sympathetic nervous system is compromised whereas the ability of cardiomyocyte to activate adrenergic and cholinergic pathways is preserved.

The lack of differences between WT and mutant mice in all these parameters strongly indicates that the gross morphological defects reported by Stankunas et al. do not contribute to the phenotype we report here. We now discuss this point in detail (Discussion, second paragraph) and completed our manuscript with a more detailed description of echographic parameters in the Results section (subsection “*Meis1* inactivation in the PNS compromises the sympatho-vagal regulation of cardiac function”, third paragraph).

*3) The authors should provide data on the innervation density in adult hearts, preferably of mice that show the described physiological phenotype to exclude the possibility that the surviving neurons have sprouted between E16.5 and adult stages.*

As mentioned by the reviewers in this comment, we understand the relevance of investigating sympathetic sprouting. A recent study indicates that sympathoectomy in neonatal rat by 6-OHDA injection leads to a moderate sympathetic sprouting between the 7^th^ day and the 8^th^ week post injection (Kreipke and Birren, 2015). To answer this point, we analyzed the innervation density in the heart of adult WT and *HtPA^CRE^/Meis1^LoxP/Lox^*^P^ mice. The whole hearts were embedded and cryo-sectioned at 14µm thickness. Every eighth section (representing 20-38 sections in total) underwent immunochemistry for TH and was fully scanned using a Nanozoomer using the 40X objective. The total length of all TH-positive fibers was measured using the ImageJ software in each section and normalized by the area of the section. Data represent the average innervation density per section. Unfortunately, due to the low level of our colony only 2 hearts for each phenotype were available in the schedule for revision. Nevertheless the results show that the innervation density in the heart of *HtPA^CRE^/Meis1^LoxP/LoxP^* mice was not increased at adult stages compared to E16.5 embryos (Figure 4) and P0 mice (Figure 4). These new results are now presented in the Figure 1—figure supplement 1. Because sympathetic sprouting following myocardial injury involves among other factors NGF signaling (Kimuraet al., 2012), this result raises the possibility again that impaired retrograde NGF in *Meis1* mutant prevents sympathetic sprouting. Another possible explanation is that *Meis1* itself is involved in sympathetic sprouting. In line with the second hypothesis, the transcription factor Stat3, a well-characterized player in axonal regeneration for peripheral neurons was also identified in our ChiP-seq experiment as a putative *Meis1* transcriptional target. Although not investigated in our study, we believe that these hypotheses are highly relevant in the field of peripheral nerves regeneration and sympathetic cardiac remodeling following injury. They would certainly deserve further attention and thorough investigations. We did not investigate these hypotheses, as we believe that it is beyond the scope of the present work.